# Exploiting multi-wavelength aerosol absorption coefficients in a multi-time resolution source apportionment study to retrieve source-dependent absorption parameters

Alice C. Forello[1], Vera Bernardoni[1], Giulia Calzolai[2], Franco Lucarelli[2], Dario Massabò[3], Silvia Nava[2], Rosaria E. Pileci[1,a], Paolo Prati[3], Sara Valentini[1], Gianluigi Valli[1], Roberta Vecchi[1,*]

[1]Department of Physics, Università degli Studi di Milano and National Institute of Nuclear Physics INFN-Milan, via Celoria 16, Milan, 20133, Italy

[2]Department of Physics and Astronomy, Università di Firenze and National Institute of Nuclear Physics INFN-Florence, via G. Sansone 1, Sesto Fiorentino, 50019, Italy

[3]Department of Physics, Università degli Studi di Genova and National Institute of Nuclear Physics INFN- Genoa, via Dodecaneso 33, Genoa, 16146, Italy

[a]now at: Laboratory of Atmospheric Chemistry (LAC), Paul Scherrer Institut (PSI), Forschungsstrasse 111, Villigen, 5232, Switzerland

*Correspondence to*: Roberta Vecchi (roberta.vecchi@unimi.it)

**Abstract.** In this paper, a new methodology coupling aerosol optical and chemical parameters in the same source apportionment study is reported. In addition to results on sources assessment, this approach gives relevant information such as estimates for the atmospheric Absorption Ångström Exponent ($\alpha$) of the sources and Mass Absorption Cross section (MAC) for fossil fuel emissions at different wavelengths.

A multi-time resolution source apportionment study using Multilinear Engine ME-2 was performed on a PM10 dataset with different time resolution (24 hours, 12 hours, and 1 hour) collected during two different seasons in Milan (Italy) in 2016. Samples were optically analysed to retrieve the aerosol absorption coefficient $b_{ap}$ (in Mm$^{-1}$) at four wavelengths ($\lambda$=405 nm, 532 nm, 635 nm and 780 nm) and chemically characterised for elements, ions, levoglucosan, and carbonaceous components. Time-resolved chemically speciated data were joined to $b_{ap}$ multi-wavelength measurements and used as input data in the multi-time resolution receptor model; this approach was proven to strengthen the identification of sources being particularly useful when important chemical markers (e.g. levoglucosan, elemental carbon, …) are not available. The final solution consisted in 8 factors (nitrate, sulphate, resuspended dust, biomass burning,

construction works, traffic, industry, aged sea salt); the implemented constraints led to a better physical description of
factors and the bootstrap analysis supported the goodness of the solution. As for $b_{ap}$ apportionment, consistently to what
expected, the two factors assigned to biomass burning and traffic were the main contributors to aerosol absorption in
atmosphere. A relevant feature of the approach proposed in this work is the possibility of retrieving many other
information about optical parameters; for example, opposite to the more traditional approach used by optical source
apportionment models, here we obtained the atmospheric Absorption Ångström Exponent (α) of the sources (α biomass
burning = 1.83 and α fossil fuels = 0.80), without any a priori assumption. In addition, an estimate for the Mass Absorption
Cross section (MAC) for fossil fuel emissions at four wavelengths was obtained and found to be consistent with literature
ranges.
It is worth noting that the approach here presented can be also applied using widespread receptor models (e.g. EPA PMF
instead of multi-time resolution ME-2) if the dataset comprises variables with the same time resolution as well as optical
data retrieved by commercial instrumentation (e.g. an Aethalometer instead of home-made instrumentation).

**1. Introduction**
Atmospheric aerosol impacts both on local and global scale causing adverse health effects (Pope and Dockery, 2006),
decreasing visibility (Watson, 2002) and influencing the climate (IPCC, 2013). To face these issues an accurate
knowledge of aerosol emission sources is mandatory.
At the state of the art, multivariate receptor models are considered a robust approach (Belis et al., 2015) to perform source
apportionment studies and the Positive Matrix Factorization (PMF) (Paatero and Tapper, 1994) has become one of the
most widely used receptor models (Hopke, 2016) in the aerosol community. In the late 1990s the Multilinear Engine
(ME-2) was developed and proved to be a very flexible algorithm to solve multilinear and quasi-multilinear problems
(Paatero, 1999). The scripting feature of this algorithm allows the implementation of advanced receptor modelling
approaches; one example is the multi-time resolution model, developed for the first time by Zhou et al. (2004), which
uses each experimental data in its original time schedule as model input. Source apportionment studies carried out by
multi-time resolution model are still scarce in the literature (Zhou et al., 2004; Ogulei et al., 2005; Kuo et al., 2014; Liao
et al., 2015; Crespi et al., 2016; Sofowote et al., 2018) although this methodology is very useful in measurement
campaigns when instrumentation with different time resolution (minutes, hours or days) is available as high time
resolution data can be exploited without averaging them over the longest sampling interval.
It is noteworthy that the combination of time-resolved chemically speciated data with the information obtained from
instrumentation measuring aerosol optical properties at different wavelengths (e.g. the absorption coefficient $b_{ap}$) is
suggested as one of the future investigations of receptor modelling (Hopke, 2016); however, to the best of our knowledge,
very few attempts in this direction have been done (e.g. Peré-Trepat et al., 2007; Xie et al., 2019). Wang et al. (2011,
2012) introduced in a source apportionment study the Delta-C (Delta-C = BC@370 nm − BC@880 nm from Aethalometer
measurements) as an input variable and found that Delta-C was very useful in separating traffic from biomass burning
source contributions.
The wavelength dependence of the aerosol absorption coefficient ($b_{ap}$) can be empirically considered proportional to $\lambda^{-\alpha}$,
where $\alpha$ is the Absorption Ångström Exponent; $\alpha$ depends on particles composition and size, and it is a useful parameter
to gain information about particles type in atmosphere (see e.g. Yang et al., 2009). Among PM components, black carbon
(BC) is the main responsible for light absorption in atmosphere; in fact, it is considered the main PM contributor to global
warming and the second most important anthropogenic contributor after $CO_2$ (Bond et al., 2013). Black carbon refers to
a fraction of the carbonaceous aerosol that shares peculiar features about microstructure, morphology, thermal stability,
solubility, and light absorption (Petzold et al., 2013); in particular, it is characterised by a wavelength-independent
imaginary part of the refractive index over visible and near-visible regions. In the last decade, experimental studies
evidenced also the role of another absorbing component i.e. brown carbon (BrC), referred to as light-absorbing organic
matter of various origins with increasing absorption towards lower wavelengths, especially in the UV region (Andreae
and Gelencsér, 2006). BrC is an aerosol component that also affects the elemental vs. organic carbon correct separation
when using thermal-optical methods as recently outlined by Massabò et al. (2016).
Source apportionment optical models based only on multi-wavelength measurements of $b_{ap}$ are available in the literature,
i.e. the widespread Aethalometer model (Sandradewi et al., 2008a) and the more recent Multi-Wavelength Absorption
Analyzer (MWAA) model (Massabò et al., 2015; Bernardoni et al., 2017b). Briefly, these models allow to estimate the
contribution of sources to aerosol absorption in atmosphere exploiting their different dependence on $\lambda$ (different $\alpha$). As a
step forward, MWAA provides the $b_{ap}$ apportionment in relation to both the sources and the components (i.e. BC and
BrC) and gives also an estimate for $\alpha$ of BrC. Source apportionment optical models usually assume two contributors to
$b_{ap}$, namely fossil fuels combustion and biomass burning (only few exceptions are present in the literature, e.g. Fialho et
al., 2005). In most cases this assumption is well founded, except in presence of episodic events that give a not negligible
contribution to aerosol absorption in atmosphere, such as the transport of mineral dust from the Saharan desert (Fuzzi et
al., 2015). Moreover, the above-mentioned models need a priori assumption about $\alpha$ values of the sources; this is the most
critical step, since $\alpha$ depends on the kind of fuel, burning conditions and aging processes in the atmosphere and wide
ranges for $\alpha$ are reported in literature (e.g. Sandradewi et al., 2008a). Without accurate determination of source-specific
atmospheric $\alpha$ (for example exploiting the information derived from source apportionment using [14]C measurements), the
applicability of models based on optical measurements is questionable (Bernardoni et al., 2017b; Massabò et al., 2015;
Zotter et al., 2017). Moreover, the generally accepted assumption of $\alpha$=1 for fossil fuels and BC, that is derived from the
theory of absorption of spherical particles in the Rayleigh regime (Seinfeld and Pandis, 2006), might not always be valid
in atmosphere due to aerosol aging processes (Liu et al., 2018).
In the frame of a source apportionment study based on multi-time resolution receptor modelling, in this work optical and
chemical datasets were joined to explore the possibility of retrieving a multi-$\lambda$ apportionment of $b_{ap}$ with no need of a-
priori assumptions on the contributing sources. Opposite, with this approach source-dependent $\alpha$ values were provided
as output. Moreover, the multi-$\lambda$ apportionment of $b_{ap}$ in each source allowed to estimate MAC values at different
wavelengths, exploiting the well-known relation $EBC=b_{ap}(\lambda)/MAC(\lambda)$ (Bond and Bergstrom, 2006) and considering the
apportioned concentrations of elemental carbon (EC) as a proxy for BC. The evaluation of atmospheric MAC values is
also not trivial due to the possible presence of absorbing components different from BC (e.g. contribution from BrC,
especially at lower wavelengths).
The original approach proposed in this work shows that coupling the chemical and optical information in a receptor
modelling process is particularly advantageous because: (1) strengthens the source identification, that is particularly
useful when relevant chemical tracers (e.g. levoglucosan, EC, …) are not available; (2) gives estimates for source-specific
atmospheric Absorption Ångström Exponent ($\alpha$) which are typically assumed a-priori in optical apportionment models;
(3) assesses MAC values at different wavelengths for specific sources.
It is also worth noting that the approach here presented is of general interest as (1) in this work optical data were retrieved
by a home-made multi-wavelength polar photometer but the same methodology could be applied to datasets combining
aerosol chemical and optical data obtained by widespread instrumentation (e.g. Aethalometers for optical data); (2) input
data to the receptor model not necessarily should comprise variables acquired with different time resolution as we did
here.

**2. Material and methods**
*2.1 Site description and aerosol sampling*
Two measurement campaigns were performed during summertime (June-July) and wintertime (November-December)
2016 in Milan (Italy). Milan is the largest city (more than 1 million inhabitants, doubled by commuters everyday) of the
Po Valley, a very well-known hot-spot pollution area in Europe due to both large emissions from a variety of sources (i.e.
traffic, industry, domestic heating, energy production plants, and agriculture) and low atmospheric dispersion conditions
(e.g. Vecchi et al., 2007 and 2019; Perrone et al., 2012; Bigi and Ghermandi, 2014; Perrino et al., 2014).
The sampling site is representative of the urban background and it is situated at about 10 meters above the ground, on the
roof of the Physics Department of the University of Milan, less than 4 km far from the city centre (Vecchi et al., 2009).
It is important to note that during the sampling campaigns, a large building site was in activity next to the monitoring
station.
Aerosol sampling was carried out using instrumentation with different time-resolution. Low time resolution PM10 data,
with a sampling duration of 24 and 12 hours during summertime (20 June-22 July 2016) and wintertime (21 November-
22 December 2016), respectively, were collected in parallel on PTFE (Whatman, 47 mm diameter) and pre-fired (700 °C,
1 hour) quartz-fibre (Pall, 2500QAO-UP, 47 mm diameter) filters. Low volume samplers with EPA PM10 inlet operating
at $1 \text{ m}^3 \text{ h}^{-1}$ were used. High time resolution data were collected during shorter periods (11 July-18 July and 21 November-
28 November 2016) by a streaker sampler (D'Alessandro et al., 2003). Shortly, the streaker sampler collects the fine and
coarse PM fractions (particles with aerodynamic diameter $d_{ae} < 2.5$ μm, and $2.5 < d_{ae} < 10$ μm, respectively) with hourly
resolution. Particles with $d_{ae} > 10$ μm impact on the first stage and are discarded; the coarse fraction deposits on the second
stage, consisting of a Kapton foil; finally, the fine fraction is collected on a polycarbonate filter. The two collecting
supports are kept in rotation with an angular speed of about $1.8° \text{ h}^{-1}$ to produce a circular continuous deposit on both
stages.
Meteorological data were available at a monitoring station belonging to the regional environmental agency (ARPA
Lombardia) which is less than 1 km far away.

*2.2 PM mass concentration and chemical characterisation*
In this Section, chemical analyses performed on samples are summarised. As measured concentration in each sample was
characterised by its own uncertainty, only ranges for experimental uncertainties and minimum detection limits (MDLs)
for every set of variables are reported.
PM10 mass concentration was determined on PTFE filters by gravimetric technique. Weighing was performed by an
analytical balance (Mettler, model UMT5, 1 μg sensitivity) after a 24 hours conditioning period in an air-controlled room
as for temperature ($20 \pm 1$ °C) and relative humidity ($50 \pm 3$ %) (Vecchi et al., 2004).
These filters were then analysed by Energy Dispersive X-Ray Fluorescence (ED-XRF) analysis to obtain the elemental
composition (details on the procedure can be found in Vecchi et al., 2004). For most elements and samples, concentrations
were characterised by relative uncertainties in the range 7-20 % (higher uncertainties for elements with concentrations
next to MDLs) and minimum detection limits of $0.9\text{-}30 \text{ ng m}^{-3}$ with the above mentioned sampling conditions.
For each quartz-fibre filter, one punch ($1.5 \text{ cm}^2$) was extracted by sonication (1 h) using 5 ml ultrapure Milli-Q water;
this extract was analysed to measure both levoglucosan and inorganic anions concentrations. Levoglucosan concentration
was determined by High-Performance Anion Exchange Chromatography coupled with Pulsed Amperometric Detection
(HPAEC-PAD) (Piazzalunga et al., 2010) only in winter samples. Indeed, as already pointed out by other studies at the

same sampling site (Bernardoni et al., 2011) and as routinely measured at monitoring stations in Milan by the Regional Environmental Agency (private communication), levoglucosan concentrations during summertime are lower than the MDL of the technique (about 6 ng m$^{-3}$), due to both lower emissions (no influence of residential heating and negligible impact from other sources) and higher OH levels in the atmosphere depleting molecular markers concentrations (Robinson et al., 2006; Hennigan et al., 2010). Uncertainties on levoglucosan concentration were about 11 %. The measurement of main water-soluble inorganic anions (SO$_4^{2-}$ and NO$_3^-$) was performed by Ion Chromatography (IC); these data had MDL of 25 and 50 ng m$^{-3}$ with summertime and wintertime sampling conditions, respectively, and uncertainties of about 10 %. Unfortunately, due to technical problems no data on ammonium were available. Details on the analytical procedure for IC analysis are reported in Piazzalunga et al. (2013).

Another punch (1.0 cm$^2$) of each quartz-fibre filter was analysed by Thermal Optical Trasmittance analysis (TOT, Sunset Inc., NIOSH-870 protocol) (Piazzalunga et al., 2011) in order to assess organic and elemental carbon (OC and EC) concentrations. MDL was 75 and 150 ng m$^{-3}$ with summertime and wintertime sampling conditions, respectively, and uncertainties were in the range 10-15 %.

Hourly elemental composition was assessed by Particle Induced X-ray Emission (PIXE) technique, using a properly collimated proton beam and scanning the deposits in steps corresponding to 1-hour aerosol deposit (details in Calzolai et al., 2015). In this work, fine and coarse elemental concentrations determined by PIXE analysis were added up to obtain PM10 concentrations with hourly resolution as low time resolution PM10 samples were also available. PM10 hourly concentrations of most elements and samples were characterised by relative uncertainties in the range 10-30 % (higher uncertainties for elements near MDL) and MDLs ranged from a minimum of 0.1 to a maximum of 15 ng m$^{-3}$ (higher MDLs typically detected for Z<20 elements).

*2.3 Aerosol light-absorption coefficient measurements*

The aerosol absorption coefficient (b$_{ap}$) at the 4 wavelengths $\lambda$ = 405, 532, 635 and 780 nm was measured on both low and high time resolution samples with the home-made polar photometer PP_UniMI (Vecchi et al., 2014; Bernardoni et al., 2017c). Results on b$_{ap}$ obtained by this custom photometer resulted in very good agreement against multi-angle absorption photometer (MAAP) data at 635 nm (Vecchi et al., 2014; Bernardoni et al., 2017c). More recently, in the frame of a collaboration with the Jülich Forschungszentrum (Germany), the Absorption Ångström Exponents retrieved by extinction minus scattering measurements were compared at two wavelengths (630 nm and 450 nm) with the one obtained by PP_UniMI data for laboratory-generated aerosols. The agreement with Cabot soot was in general very good as for both b$_{ap}$ at two wavelengths and Absorption Ångström Exponent estimates, i.e. comparability within one standard deviation (data not yet published, preliminary results reported in Valentini et al., 2019).

Low time resolution optical measurements taken into account were those performed on PTFE filters since their physical characteristics can be considered more similar to polycarbonate filters used by the streaker sampler. Moreover, previous works reported a bias on $b_{ap}$ measured by instrumentation using fibre filters (e.g. Cappa et al., 2008: Lack et al., 2008; Davies et al., 2019; and references therein); Vecchi et al. (2014) quantified in about 40 % the effect caused in $b_{ap}$ values (assessed at 635 nm) by sampling artefacts due to organics in aerosol samples collected in Milan when comparing aerosol samples collected in parallel quartz-fibre and PTFE filters.

For high time resolution samples, $b_{ap}$ was measured only in the fine fraction collected on polycarbonate filters, since absorption from the Kapton foil on which the coarse fraction was collected did not allow $b_{ap}$ assessment. Anyway, $b_{ap}$ values in PM2.5 and PM10 were expected to be fairly comparable, as most of the contribution to aerosol absorption in atmosphere is typically given by particles in the fine fraction at heavily polluted urban sites like Milan. To verify this assumption, high time resolution $b_{ap}$ data in PM2.5 were averaged on the time scale of low time resolution $b_{ap}$ in PM10 for comparison. They turned out to be in good agreement, between 11 % and 13 % depending on the $\lambda$, except for $b_{ap}$ at $\lambda$=405 nm that showed a higher difference (27 %) but with most data (83 %) within experimental uncertainties. To take into account for this difference, $b_{ap}$ data at $\lambda$=405 nm were homogenised before their insertion into the model, following the criterion used for chemical species (for further detail about homogenisation procedure, see Sect. 2.4 and Sect. 2.5).

Uncertainties on $b_{ap}$ were estimated as 15 % and MDL was in the range 1-10 Mm$^{-1}$ depending on sampling duration and wavelength as already reported in our previous works (Vecchi et al., 2014; Bernardoni et al., 2017c). Experimental uncertainties and MDL of optical absorption data were used as a starting point to estimate the uncertainties introduced in the model. Pre-treatment procedure for these data was the same used for chemical variables (see also Sect. 2.5). Optical system stability was checked during the measurement session, evaluating the reproducibility of the measurement on a blank test filter. Laser stability was also checked at least twice a day and the recorded intensities were used to normalise blank and sampled filters analysis.

*2.4 Model description*

Multivariate receptor models (Henry, 1997) are among the most widespread and robust approaches used to carry out source apportionment studies for atmospheric aerosol (Belis et al., 2014 and 2015). In particular, the Positive Matrix Factorization (Paatero and Tapper, 1994; Paatero, 1997) had been extensively used in the literature and, afterwards, the Multilinear Engine ME2 (Paatero, 1999 and 2000) introduced the possibility of solving all kinds of multilinear and quasi-multilinear problems. The fundamental principle of these modelling approaches is the mass conservation between the emission source and the receptor site; using the information carried by aerosol chemical composition assessed on a number of samples collected at the receptor site, a mass balance analysis can be performed to identify the factors influencing

aerosol mass concentrations (Hopke, 2016). Factors can be subsequently interpreted as the main sources impacting the
site, exploiting knowledge about the most relevant sources in the investigated area and the adoption of fingerprints
available from previous literature works (Belis et al., 2014). Referring to the input data as matrix X (matrix elements $x_{ij}$),
the chemical profile of the factors as matrix F (matrix elements $f_{kj}$), and the time contribution of the factors as matrix G
(matrix elements $g_{ik}$), the main equation of a bilinear problem can be written as follows:
$$x_{ij} = \sum_{k=1}^{P} g_{ik}f_{kj} + e_{ij} \qquad (1)$$

where the indices i, j, and k indicate the sample, the species, and the factor, respectively; P is the number of factors and
the matrix E (matrix elements $e_{ij}$) is composed by the residuals, i.e. the difference between measured and modelled values.
In this way, a system of NxM equations is established, where N is the number of samples and M is the number of species.
The solution of the problem is computed minimising the object function Q defined as:
$$Q = \sum_{i=1}^{N} \sum_{j=1}^{M} \left(\frac{e_{ij}}{\sigma_{ij}}\right)^2 \qquad (2)$$

where $\sigma_{ij}$ are the uncertainties related to the input data.
The multi-time resolution receptor model was developed in order to use each data value in its original time schedule,
without averaging the high time resolution data or interpolating the low time resolution data (Zhou et al., 2004; Ogulei et
al., 2005). The main Eq. (1) is consequently modified as below:
$$x_{sj} = \frac{1}{t_{s2}-t_{s1}+1}\sum_{k=1}^{P} f_{kj} \sum_{i=t_{s1}}^{t_{s2}} g_{ik}\eta_{jm} + e_{sj} \qquad (3)$$

where the indices s, j, and k indicate the sample, the species and the factor respectively; P is the number of factors; $t_{s1}$ and
$t_{s2}$ are the starting and ending time for the s-th sample in time units (i.e. the shortest sampling interval, that is 1 hour for
the dataset used here) and i represents one of the time units of the s-th sample. $\eta_{jm}$ are adjustment factors for chemical
species replicated with different time resolution and measured with different analytical methods (represented by the
subscript m).
If $\eta$ is close to unity, species concentration measured by different analytical approaches can be considered in good
agreement; non-replicated species have adjustment factors set to unity by default. In this work, the adjustment factors
were always set to unity in the model; to take into account the use of different aerosol samplers (i.e. low volume sampler
with EPA inlet and streaker sampler) and different analytical techniques to obtain the elemental composition (i.e. ED-
XRF and PIXE), concentrations of replicated species with different time resolution were homogenised before inserting
them into the input matrix X, as will be explained in Sect. 2.5. Applying this data treatment procedure, it is possible to
avoid to check if the $\eta$ values calculated by the model are consistent with differences in experimental data characterised
by high and low time resolution. Otherwise, this step should always be performed after running the model.
In the multi-time resolution model a regularisation equation is introduced, since some sources could contain few or no
species measured with high time resolution:

$$g_{(i+1)\,k} - g_{ik} = 0 + \varepsilon_i \qquad (4)$$

where $\varepsilon_i$ represent the residuals.
As already pointed out by Ogulei et al. (2005), a weighing parameter for low resolution species might be necessary; in
this study, it was implemented in the equations and set at 0.5 for strong species (not applied to weaker species as Na, Mg,
and Cr, see Sect. 2.5) in 24-h or 12-h samples.
Equations (3) and (4) are solved using the Multilinear Engine (ME) program (Paatero, 1999). In Eq. (2), the object
function Q takes into account residuals from the main Eq. (3) and from the auxiliary equations (regularisation Eq. (4),
normalisation equation, pulling equations, and constraints).
In this work, the multi-time resolution model implemented by Crespi et al. (2016) was used; therefore, constraints were
inserted in the model and the bootstrap analysis was also performed to evaluate the robustness of the final solution.

*2.5 Input data*
As already mentioned in Sect. 2.4, instead of using adjustment factors in the model (all set equal to one), concentrations
of replicated species with different time resolution were pre-homogenised and then inserted into the input matrix X.
Concentration data with longer sampling interval (24 and 12 hours in this work) were considered as benchmark, since
analytical techniques usually show a better accuracy on concentration values far from MDL (i.e. samples collected on
longer time intervals) (Zhou et al., 2004; Ogulei et al., 2005).
Variables were then classified as weak and strong according to the signal-to-noise ratio (S/N) criterion (Paatero, 2015).
For hourly data only strong variables (S/N ≥ 1.2) were considered; for low time resolution data also weaker variables as
Na, Mg and Cr (with S/N equal to about 0.8), that resulted strong variables in hourly samples, were also included although
under-weighed (i.e. associated uncertainties comparable to concentration values) in order to avoid the exclusion of too
many data. Indeed, excluding these low time resolution variables from the analysis gave rise to artificial high values in
the time contribution matrix for sources traced by these species (in this case it was particularly important for aged sea salt
traced by Na and Mg, see Sect. 3.2); this oddity was already reported by Zhou et al. (2004).
Every measured variable in each sample is characterised by its own uncertainty; ranges of experimental uncertainties and
MDLs are reported in Sect. 2.2 and 2.3 for chemical and optical analyses, respectively. Variables with more than 20 %

of the concentration data below MDL values were omitted from the analysis (Ogulei et al., 2005). The procedure described in Polissar et al. (1998) was followed to treat uncertainties and below MDL data, starting from experimental uncertainties and MDLs. In general, missing concentration values were estimated by linear interpolation of the measured data and their uncertainties were assumed as three times this estimated value (Zhou et al., 2004; Ogulei et al., 2005). As for summertime levoglucosan data (not available), the approach was to include them as below MDL data and not as missing data following Zhou et al. (2004), who underlined that the multi-time resolution model is more sensitive to missing values than the original PMF model. In order to avoid double counting, in this study S was chosen as input variable instead of $SO_4^{2-}$ as it was determined on both low time and high time resolution samples (by XRF and PIXE analysis, respectively, see Calzolai et al., 2008). However, elemental $SO_4^{2-}$ and S concentrations showed a high correlation (correlation coefficient R=0.98) and the Deming regression gave a slope of $2.69 \pm 0.13$ (sulphate vs. sulphur) with an intercept of $-198 \pm 82$, i.e. compatible with zero within 3 standard deviations. The slight difference (of the order of 10%) between the estimated slope and the $SO_4^{2-}$-to-S stoichiometric coefficient (i.e. 3) can be ascribed to either a small fraction of insoluble sulphate or to the use of different analytical techniques.

PM10 mass concentrations were included in the model with uncertainties set at four times their values (Kim et al., 2003). In the end, 22 low time resolution variables (PM10 mass, Na, Mg, Al, Si, S, K, Ca, Cr, Mn, Fe, Cu, Zn, Pb, EC, OC, levoglucosan, $NO_3^-$, $b_{ap}$ 405nm, $b_{ap}$ 532nm, $b_{ap}$ 635nm, $b_{ap}$ 780nm) and 17 hourly variables (Na, Mg, Al, Si, S, K, Ca, Cr, Mn, Fe, Cu, Zn, Pb, $b_{ap}$ 405nm, $b_{ap}$ 532nm, $b_{ap}$ 635nm, $b_{ap}$ 780nm) were considered.

The input matrix X consisted in 386 samples and the total number of time units was 1117. The analysis was performed in the robust mode; lower limit for G contribution was set to -0.2 (Brown et al., 2015) and the error model em=-14 was used for the main equation with $C_1$= input error, $C_2$= 0.0 and $C_3$=0.1 (Paatero, 2012) for both chemical and optical absorption data.

Sensitivity tests on the uncertainty of absorption data were performed starting from a minimum uncertainty of 10 %. Lower uncertainties were considered not physically meaningful from an experimental point of view. ME-2 analyses performed with 10 % uncertainty on absorption data gave very similar results to the base case solution presented in the Supplement (Figure S1 and Table S3), with no differences in mass apportionment and a maximum variation in the concentrations of chemical and optical profiles (matrix F) of 7 % when considering significant variables in each profile (i.e. EVF higher or near 0.30). Opposite, considering an uncertainty of 20 % on absorption data, the solution significantly differed from the base case one presented in the Supplement and showed less physical meaning. Indeed, the factors assigned to resuspended dust and construction works got mixed, and a new unique factor (traced almost exclusively by Pb) appeared, with mass contribution equal to zero. Thus, the estimated relative uncertainty of 15 % was here considered appropriate for optical variables.

It is also noteworthy that ME-2/PMF analysis is not a-priori harmed by the use of joint matrices containing different units
(see e.g. Paatero, 2018). Indeed, if different units are present in different columns of matrix X, the output data in factor
matrix G are pure numbers and elements in a column of factor matrix F carry the same dimension and unit as the original
data in matrix X. In addition, as we did in this work, the average total contribution to the mass of a specific source due to
species in a certain factor in matrix F must be retrieved a-posteriori summing up only mass contributions by chemical
components (i.e. excluding optical components in matrix F).
To the authors' knowledge, this is the first time that the absorption coefficients at different wavelengths were introduced
in the multi-time resolution model and used to more robustly identify the sources; moreover, the optical information was
also exploited to retrieve additional information such as the Absorption Ångström Exponent ($\alpha$) of the sources and MAC
values in an original way.

**3. Results and discussion**
*3.1 Concentration values*
In Table S1 (Supplement) basic statistics on mass and chemical species concentrations at different time resolution are
given.
Most variables showed higher mean and median concentrations during the winter campaign, when atmospheric stability
conditions influenced the monitoring site; exceptions were Al, Si and Ca which had lower median concentrations (as
detected in low time resolution samples). This was not unexpected as they are typical tracers of soil dust resuspension
(Viana et al., 2008) that can be more relevant during summertime due to drier soil conditions and higher atmospheric
turbulence. Moreover, the good correlation between these elements (Al vs Si: $R^2$=0.94 and Ca vs Si: $R^2$=0.78) suggested
the common origin.
Potassium was the element showing the most different median concentrations in the two seasons; its median concentration
in low time resolution samples was 284 ng m$^{-3}$ (10th-90th percentile: 151-344 ng m$^{-3}$) and 660 ng m$^{-3}$ (10th-90th percentile:
349-982 ng m$^{-3}$) in summer and winter, respectively. K is an ambiguous tracer, since it is emitted by a variety of sources
such as crustal resuspension and biomass burning. In our dataset, wintertime K values showed a good correlation with
levoglucosan concentrations ($R^2$=0.71) suggesting an impact of biomass burning as levoglucosan is a well-known tracer
for biomass burning emissions in winter samples (Simoneit al., 1999). Also looking at K-to-Si ratio (the latter taken as
soil dust marker) significant seasonal differences came out; it was 0.35 ± 0.15 in high time resolution summer samples
and 2.0 ± 2.2 in winter ones, to be compared with the much more stable ratio for Al/Si (i.e. 0.26 ± 0.04 and 0.28 ± 0.09
in summer and winter, respectively).
Among the elements typically associated to anthropogenic sources, Fe and Cu showed a good correlation (e.g. $R^2$=0.72
on hourly resolution samples) as well as Cu and EC (Cu vs EC: $R^2$=0.84, on low time resolution data). In addition, the
diurnal pattern of Fe and Cu showed traffic rush-hours peaks (7-9 a.m. and around 19 p.m. as shown in Fig.1). These
results were suggestive of a common source. Indeed, these aerosol chemical components are reported in the literature as
tracers for vehicular emissions (e.g. Viana et al., 2008; Thorpe and Harrison, 2008).

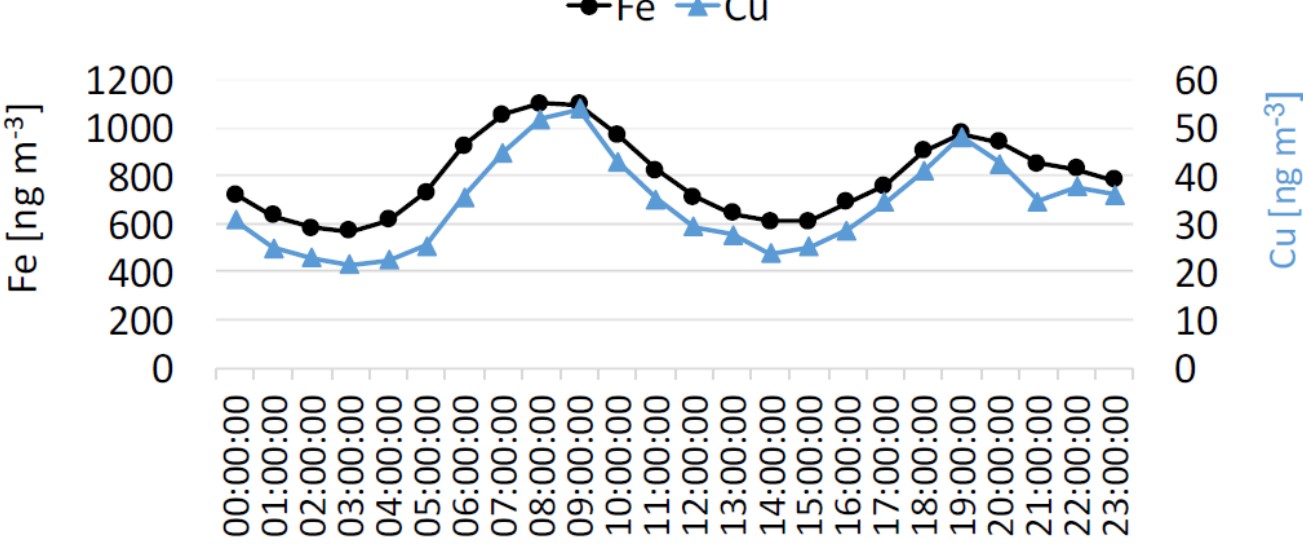


Figure 1: Diurnal profile of Fe and Cu concentrations (in ng m$^{-3}$).

In Table S2 (Supplement) also basic statistics on $b_{ap}$ values referred to low resolution samples collected on PTFE are
reported. Diurnal mean temporal patterns for $b_{ap}$ at different wavelengths (retrieved from hourly resolved data) are
displayed in Fig. 2.

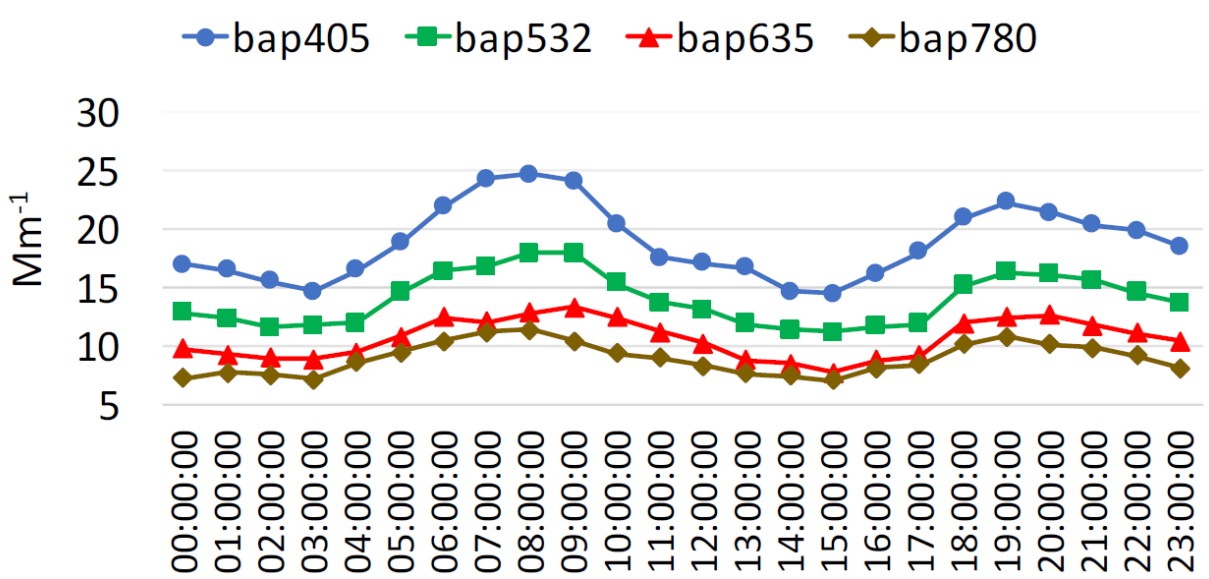


Figure 2: Diurnal profile of aerosol absorption coefficient measured at different wavelengths.

*3.2 Source apportionment with multi-time resolution model*

Different number of factors (5-10) were explored; after 30 convergent runs, the 8-factor base-case solution corresponding to the lowest Q value (2086.88) was firstly selected (see Fig. S1 in the Supplement). It is important to notice that the model was run using all variables (chemical + optical) as explained in Sect. 2.5. A lower or higher number of factors caused ambiguous chemical profiles and the physical interpretation singled out clearly mixed sources for a lower number of factors or unique factors in case of more factors (i.e. Pb for 9 factors); moreover, inconsistent mass closure was detected increasing the number of factors (e.g. the sum of species contribution was up to 25 % higher than the mass for the 10-factor solution). In the 8-factor base case solution, the mass was well reconstructed by the model ($R^2$=0.98), with a slope of $0.98 \pm 0.02$ and negligible intercept=$0.51 \pm 0.89$ $\mu$g m$^{-3}$.

The factor-to-source assignment process was based on both the Explained Variation for F matrix (EVF) values - which are typically higher for chemical tracers (Lee et al., 1999; Paatero, 2010) - and the physical consistence of factor chemical profiles. In the chosen solution, the not explained variation was lower than 0.25 for all variables. The scaled residuals showed a random distribution of negative and positive values in the $\pm$ 3 range, with a Gaussian shape for most of the variables (Fig. S2 in the Supplement).

Using EVF and chemical profiles reported in Fig. S1(a), the 8 factors were tentatively assigned to specific atmospheric aerosol sources: nitrate, sulphate, resuspended dust, biomass burning, construction works, traffic, industry, and aged sea salt. In Table S3 (in the Supplement) absolute and relative average source contributions to PM10 mass are reported.

Although the above mentioned base-case solution was a satisfactory representation of the main sources active in the area (as reported in previous works, see e.g. Marcazzan et al., 2003; Vecchi et al., 2009 and 2018; Bernardoni et al., 2011 and 2017a; Amato et al., 2016), the chemical profiles of some factors were improved exploring rotated solutions. The most relevant case was represented by aged sea-salt where typical diagnostic ratios such as Mg/Na and Ca/Na were not well reproduced (in bulk sea water equal to 0.12 and 0.04, respectively, as reported e.g. in Seinfeld and Pandis, 2006) and the chemical profile itself was too much impacted by the presence of Fe compared to bulk sea water composition. Therefore, the above-mentioned diagnostic ratios were here used as constraints and Fe was maximally pulled down in the chemical profile. The effective increase in Q was of about 61 units (Q=2147), with a percentage increase of about 3 %; as a rule of thumb, an increase in the Q value of a few tens is generally considered acceptable (Paatero and Hopke, 2009). It is noteworthy that an improvement in the chemical profiles was achieved with negligible differences compared to the base-case solution as for all other relevant features of the solution (i.e. EVF, residuals, mass reconstruction, source apportionment). Therefore, the 8-factor constrained solution was considered the most physically reliable; results are presented in Table 1 and Fig. 3 and discussed in detail in the following.

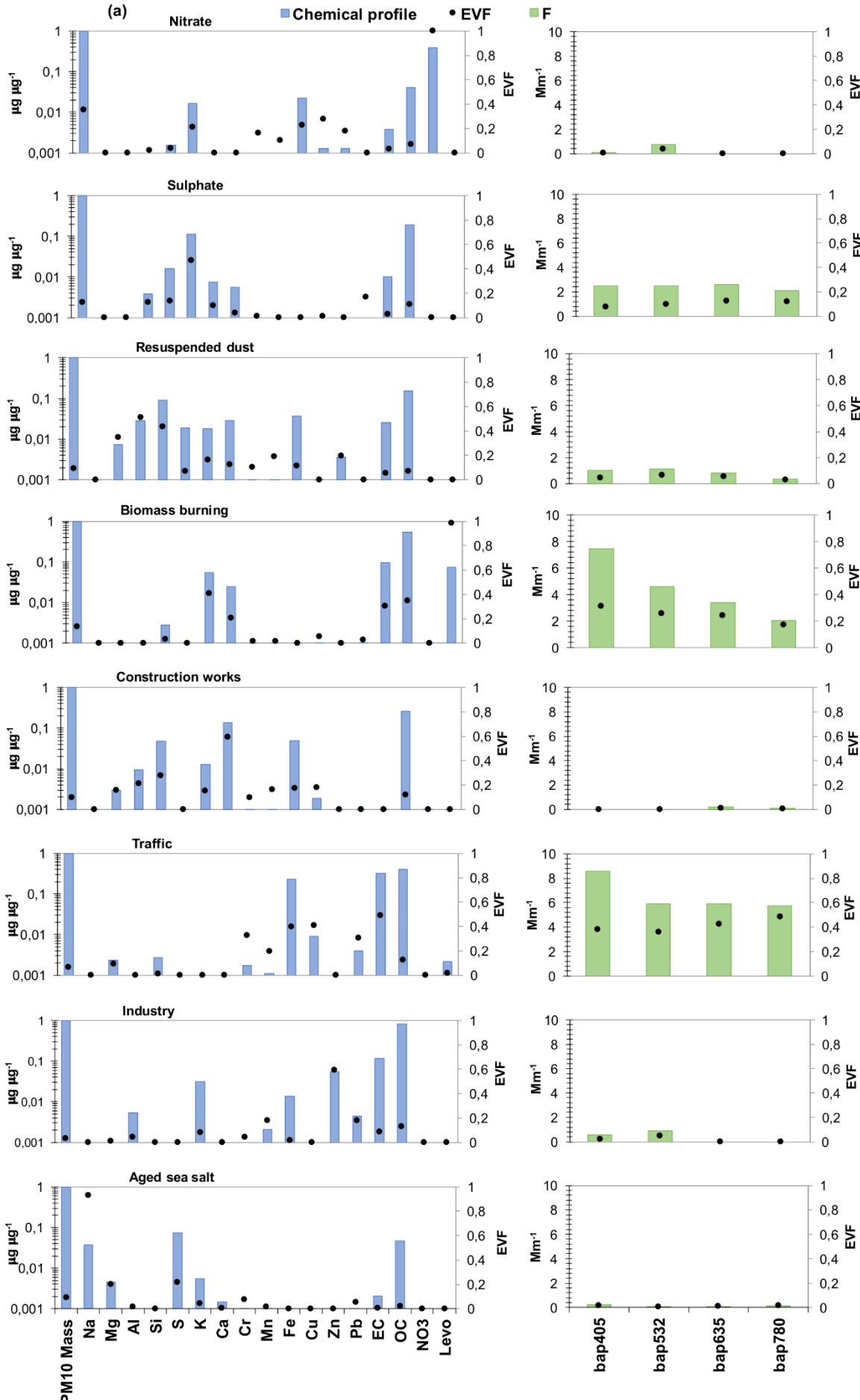

Figure 3: (a) Chemical profiles of the 8-factor constrained solution (b) $b_{ap}$ apportionment of the 8-factor constrained
solution.

| Factors | Summer [$\mu g\ m^{-3}$] | Winter [$\mu g\ m^{-3}$] | Total [$\mu g\ m^{-3}$] |
|---|---|---|---|
| Nitrate | 3.6 (15 %) | 21.1 (44 %) | 10.2 (31 %) |
| Sulphate | 6.3 (26 %) | 8.1 (17 %) | 7.0 (21 %) |
| Resuspended dust | 4.6 (19 %) | 1.7 (4 %) | 3.5 (11 %) |
| Biomass burning | 0.32 (1 %) | 8.3 (17 %) | 3.3 (10 %) |
| Construction works | 5.9 (24 %) | 3.4 (7 %) | 4.9 (15 %) |
| Traffic | 1.4 (6 %) | 2.2 (5 %) | 1.7 (5 %) |
| Industry | 0.86 (4 %) | 1.2 (3 %) | 1.0 (3 %) |
| Aged sea salt | 1.4 (6 %) | 1.8 (4 %) | 1.6 (5 %) |

Table 1: Absolute and relative average source contributions to PM10 mass in the 8-factor constrained solution.

The factor interpreted as nitrate fully accounted for the explained variation of $NO_3^-$. This factor contained a significant
fraction of nitrate in the chemical profile (39 %) and all nitrate was present only in this factor. This source was by large
the most significant one at the investigated site, explaining about 31 % of the PM10 mass over the whole campaign (a
similar estimate – 26 % - was reported by Amato et al. (2016) during the AIRUSE campaign in Milan in 2013) raising up
to 44 % during wintertime (comparable to 37 % reported by Vecchi et al. (2018)). Indeed, the Po valley is well-known
for experiencing very high nitrate concentrations during wintertime (Vecchi et al., 2018; and references therein) because
of large emissions of gaseous precursors related to urban and industrial activities, biomass burning used for residential
heating, high ammonia levels due to agricultural fields manure and – last but not the least – poor atmospheric dispersion
conditions.
The factor associated to sulphate shows EVF=0.47 for S and much lower EVF for all the other variables in the factor.
Considering the contribution of S in the chemical profile in terms of sulphate and ammonium sulphate, the relative
contribution of sulphur components in the profile increases from 11 % (S) up to 45 % (ammonium sulphate). The latter
is the main sulphur compound detected in the Po valley as reported in previous papers such as e.g. Marcazzan et al. (2001)
and was by far the highest contributor in the chemical profile. The other important contributor was OC (19 %), whose
impact on PM mass increased up to 30 % when reported as organic matter using 1.6 as the organic carbon-to-organic
matter conversion factor for this site (Vecchi et al., 2004). Due to the secondary origin of the aerosol associated to this
factor, it was not surprising to find also a significant OC contribution; indeed, aerosol chemical composition in Milan is
impacted by highly oxygenated components due to aging processes favoured by strong atmospheric stability (Vecchi et
al., 2018 and 2019). In this factor, EC contributed for about 1 %. Considering the total EC concentration reconstructed
by the model, the EC fraction related to the sulphate factor was about 6 %. Opposite to sulphates, EC has a primary origin;
however, its presence with a very similar percentage (4-5 %) in a sulphate chemical profile was previously pointed out in
Milan, indicating a more complex mixing between primary and secondary sources (Amato et al., 2016). The sulphate
factor accounted for 21 % of the PM10 mass.
The factor identified as resuspended dust is mainly characterised by high EVFs and contributions coming from Al, Si and
Mg, i.e. crustal elements. The Al/Si ratio is 0.31, very similar to the literature value for average crust composition (Mason,
1966); the relatively high contribution of OC in the chemical profile (15 %) and the presence of EC (about 2.6 %), indicate
that there is very likely a mixing with road dust (Thorpe and Harrison, 2008). This source accounts for about 11 % of the
PM10 mass.
The factor identified as biomass burning was characterised by high EVF for levoglucosan (0.98), a known tracer for this
source as it is generated by cellulose pyrolysis; EVF higher than 0.3 were also found for K, OC, and EC. In the source
chemical profile, OC contributed for 54 %, EC for 10 %, levoglucosan for 7 %, and K for 5 %. The average biomass
burning contribution during this campaign was 10 % (up to 17 % in wintertime). Anticipating the discussion presented in
detail in Sect. 3.3, it is worth noticing that the second largest contribution to the aerosol absorption coefficient after traffic
was detected in this factor.
The factor with high EVF (0.60) for Ca was associated to construction works, following literature works (e.g. Vecchi et
al., 2009; Bernardoni et al., 2011; Dall'Osto, 2013; Crilley et al., 2017; Bernardoni et al., 2017a; and references therein).
Major contributors to the chemical profile were Ca (13 %), OC (26 %), Fe, and Si (5 % each). This factor accounted on
average for 15 % to PM10 mass. As already mentioned, during the campaign a not negligible contribution from this
source was expected, due to the presence of a construction building site nearby the monitoring location.
In the factor here assigned to traffic (primary contribution), EVF larger than 0.3 characterised EC, Cu, Fe, Cr, and Pb.
The highest relative contributions in terms of mass in the chemical profile were given by OC (41 %), EC (32 %), Fe (23
%), and Cu (1 %). The lack of relevant crustal elements such as Ca and Al in the chemical profile, suggested a negligible
impact of road dust in this factor. As reported above, at our sampling site the road dust contribution was very likely mixed
to resuspended dust and further separation of these contributions was not possible. This traffic (primary) contribution
over the whole dataset accounted for 5 % of the PM10 mass with a slightly lower absolute contribution in summer (see
Table 1). This contribution is comparable to the percentage (7 %) reported by Amato et al. (2016) for exhaust traffic
emissions but it is lower than our previous estimates (Bernardoni et al., 2011; Vecchi et al., 2018), i.e. 15 % in 2006 in
PM10 and 12 % in PM1 recorded in winter 2012. However, the current estimate seems to be still reasonable when
considering the efforts done in latest years to reduce vehicles exhaust particle emissions and the fraction of secondary
nitrate to be added to account for the overall traffic impact; indeed, a significant traffic contribution due to nitrate should
be accounted for the relevant nitrogen oxides and ammonia emissions from agriculture in the region (INEMAR ARPA-
Lombardia, 2018). Unfortunately, the non-linearity of the emission-to-ambient concentration levels relationship and the

high uncertainties in emission inventories still prevent a robust estimate of this secondary contribution to total traffic exhaust emissions. In Sect. 3.3, it will be shown that traffic is the largest contributor to aerosol absorption coefficient, a result that reinforces the interpretation of this factor as a traffic emission source.

The industry factor showed high EVF for Zn (0.59) and the second highest EVF was related to Mn (0.13). Previous studies at the same sampling site identified these elements as tracers for industrial emissions (e.g. Vecchi et al., 2018; and references therein). The chemical profile resulted enriched by heavy metals and, after traffic, it was the profile with the highest share of Cr, Mn, Fe, Cu, Zn, and Pb (explaining about 8 % of the total PM10 mass in the profile). The industry contribution was not very high in the urban area of Milan, accounting for 3 % on average.

The factor interpreted as aged sea salt was characterised by high EVF of Na (0.93) and this element was - as a matter of fact - present only in this factor chemical profile. To check the physical consistency of this assignment and considering that Milan is about 120 km away from the nearest sea coast, back-trajectories coloured by the aged sea salt concentration (in ng m$^{-3}$) were calculated through the NOAA HYSPLIT trajectory model (Draxler and Hess, 1998; Stein et al., 2015; Rolph et al., 2017) and represented using the Openair software (Carslaw and Ropkins, 2012).

When marine air masses are transported to polluted sites, sea salt particles are characterised by a Cl deficit due to reactions with sulphuric and nitric acid (Seinfeld and Pandis, 2006). In this case, the factor chemical profile was expected to be enriched in sulphate and nitrate. In this work, nitrate was not present; a very rough estimate (Lee et al., 1999) gave a maximum expected contribution of 2 % (about 82 ng m$^{-3}$) of the total nitrate mass in atmosphere, that can be considered negligible in terms of mass contribution of the sources.

Temporal patterns of Cl concentrations (not inserted in the multi-time resolution analysis as being a weak variable) during episodes were exploited to further confirm the factor-to-source association. As an example, a very short event (13/07 h.16-18) singled out by the model and representing the highest sea salt contribution during summer was analysed in further detail. Before and during the sea salt event, air masses originated from south-west compatible with Ligurian sea while soon after the event, there was a rapid change of wind direction (Fig. S3, in the Supplement). These hours were characterised by an average high wind speed of $4.8 \pm 1.7$ m s$^{-1}$ (with a maximum peak of 9.5 m s$^{-1}$) compared to $1.9 \pm 1.0$ m s$^{-1}$ average wind speed recorded during the summer campaign. In addition, Cl concentration and aged sea salt pattern showed an evident temporal coincidence in peak occurrence during the event (Fig. 4), thus supporting the source identification. Moreover, during this episode only the Cl coarse fraction increased (Fig. S4, in the Supplement) and reached about 90 % of total PM10 Cl concentration; Cl/Na ratio was $0.38 \pm 0.05$, consistent with an aging of marine air masses during advection showing the typical Cl depletion due to the interaction between sea salt particles and polluted air masses (Seinfeld and Pandis, 2006).

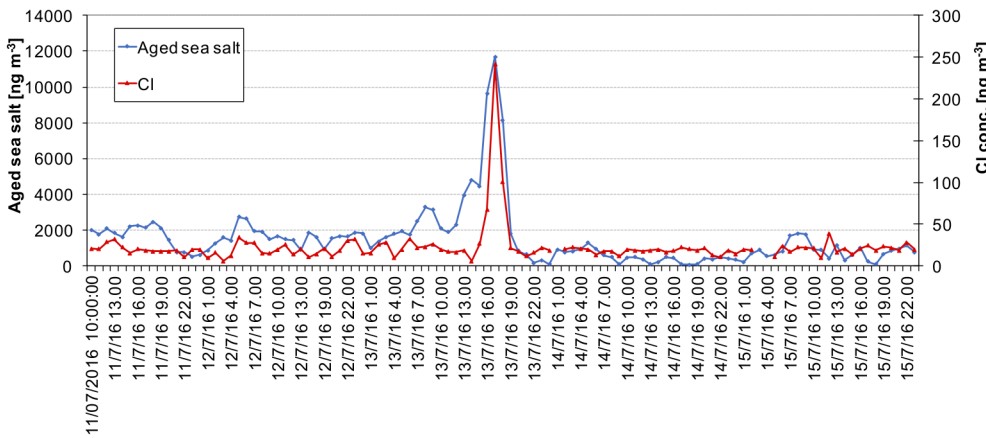


Figure 4: Temporal patterns of aged sea salt source retrieved from the multi-time resolution model and Cl concentrations
measured in atmospheric aerosol.

Bootstrap analysis was performed to evaluate the uncertainties associated to source profiles (Crespi et al., 2016). 100 runs
were carried out (see Fig. 5, values expressed in ng m$^{-3}$ or Mm$^{-1}$ on a logarithmic scale); factors were well mapped, with
Pearson coefficient always higher than 0.97, and tracers for each source showed small interquartile range, supporting the
goodness of the solution presented in this work.

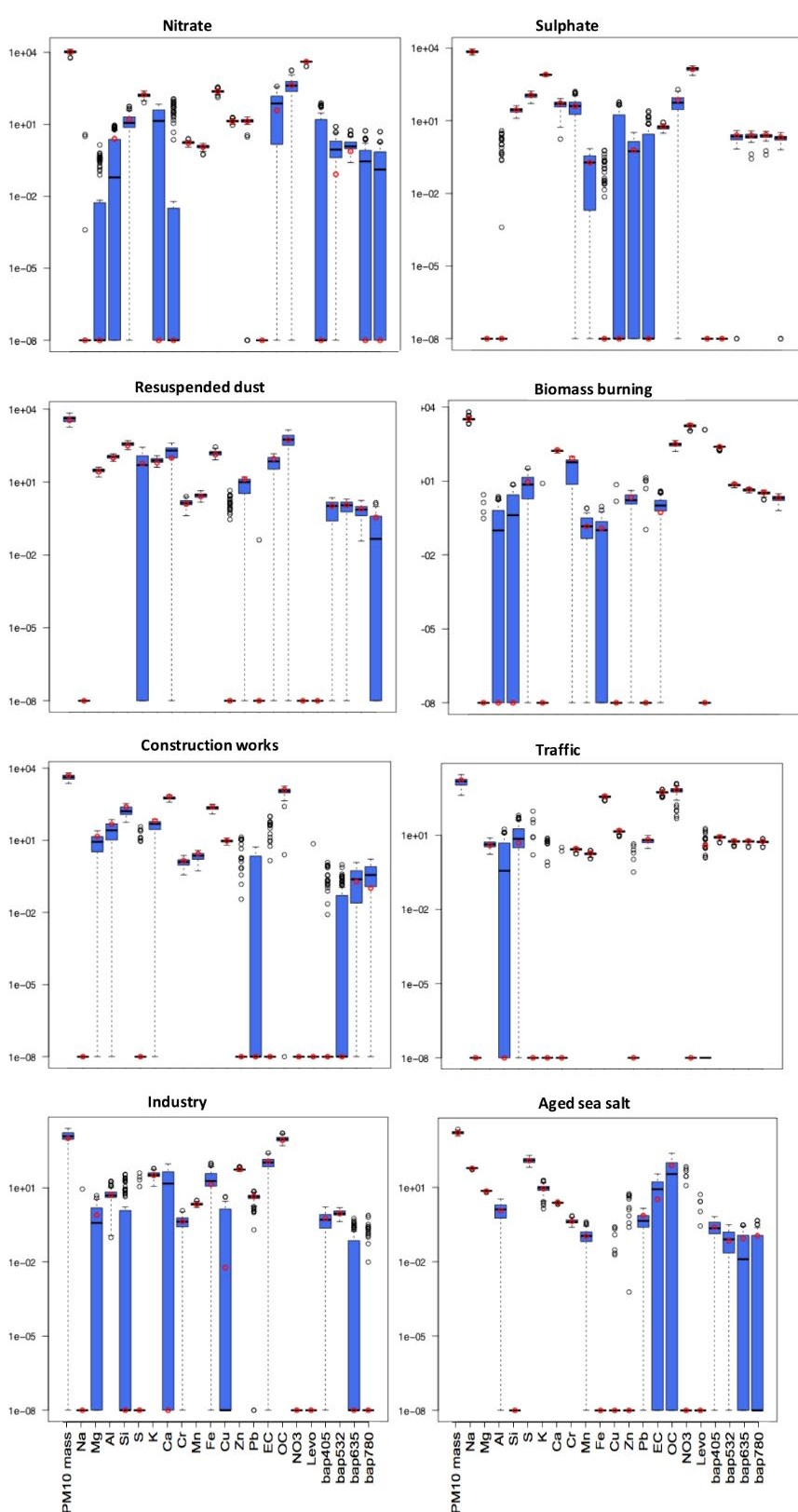


Figure 5: Box plot of the bootstrap analysis on the 8-factor constrained solution. The red dots represent the output values
of the solution of the model; the black lines the medians from the bootstrap analysis; the blue bars the 25th and 75th
percentile; the dotted lines the interval equal to 1.5 the interquartile range and the black dots the outliers from this interval.

*3.3 Improving source apportionment with optical tracers*

First of all, the use of the absorption coefficient determined at different wavelengths as input variable in the multi-time resolution model, strengthened the identification of the sources, suggesting that it can be exploited when specific chemical tracers are not available (e.g. levoglucosan for biomass burning). To prove that, a separate source apportionment study was performed with EPA PMF 5.0 (Norris et al., 2014), introducing only hourly elemental concentrations from samples collected by the streaker sampler and hourly $b_{ap}$ at different $\lambda$ measured by PP_UniMI on the same filters. Streaker samples typically lack of a complete chemical characterisation; in particular, important chemical tracers such as levoglucosan and EC are not available. In this analysis, $b_{ap}$ assessed at different wavelengths resulted particularly useful for the identification of the biomass burning factor that explained a significant percentage of the $b_{ap}$ itself (from 25 % to 35 % depending on $\lambda$) (Fig. S5, in the Supplement); without this additional information, the factor-to-source assignment would be otherwise based only on the presence of elemental potassium although it is well-known that K cannot be considered an unambiguous tracer as it is emitted by a variety of sources (see for example Pachon et al., 2013; and references therein). Furthermore, results showed that the absorption coefficient contribution was higher than 45 % in the factor labelled as traffic, highlighting the importance of exhaust emissions in a factor that would be otherwise characterised mainly on elements related to non-exhaust emissions (Cu, Fe, Cr).

From the multi-time resolution model, the two factors identified as biomass burning and traffic were the main contributors to aerosol absorption in atmosphere and showed significant EVF values. Contributions to $b_{ap}$ were 55 % and 42 % for traffic and 20 % and 36 % for biomass burning at 780 and 405 nm, respectively. The Explained Variation (EVF) of $b_{ap}$ has the maximum value at 405 nm for biomass burning (0.32) and at 780 nm for traffic (0.49), showing the tendency to decrease and increase with the wavelength, respectively.

The third contributor to aerosol absorption in atmosphere was the sulphate factor, with a contribution comparable to the biomass burning one at 780 nm (about 20 % of the total reconstructed $b_{ap}$ at this wavelength). The sulphate factor contained a small fraction of EC, as previously discussed (see Sect. 3.2). This might be explained considering that non/weakly light-absorbing material can form a coating able to enhance absorption (Bond and Bergstrom, 2006; Fuller et al., 1999) within a few days after emission (Bond et al., 2006). Laboratory experiments and simulations from in-situ measurements highlighted absorption amplification for absorbing particles coated with secondary organic aerosol (Schnaiter et al., 2003; Moffet and Prather, 2009). These processes related to particles aging can become important in the Po valley due to low atmospheric dispersion conditions and they might explain the relatively high contribution of the sulphate factor to the absorption coefficient in respect to the other sources (excluding traffic and biomass burning). Among the other sources, resuspended dust was the main contributor at all wavelengths (between 3 % and 7 % of the total

reconstructed $b_{ap}$, depending on the wavelength), likely due to the role of iron minerals. The other four sources were less
relevant in terms of EVF values and overall contributed for less than 11 %.
It is noteworthy that opposite to the approach used in source apportionment optical models, like the widespread
Aethalometer model (Sandradewi et al., 2008a) and MWAA model (Massabò et al., 2015; Bernardoni et al., 2017b), no
a-priori information about the Absorption Ångström Exponent ($\alpha$) of the fossil fuel and biomass burning sources was
introduced in the multi-time resolution model; instead, an estimate for its value was directly retrieved from the model. It
has to be mentioned that optical models are typically based on a two-source hypothesis (i.e. biomass burning and fossil
fuel emissions); an exception reported in previous works (Wang et al., 2011) concerned the use of Delta-C used as an
input variable together with chemical aerosol components in source apportionment models and proved to be very effective
in separating traffic (especially diesel) emissions from biomass combustion emissions.
Hereafter, in order to compare multi-time resolution model and optical models results, contributions due to traffic and
industry (i.e. emissions most likely connected to fossil fuel usage) were added up and labelled as "fossil fuel emissions".
Similarly to the two-source approach used in the Aethalometer model, the discussion about optical properties will be
hereafter focused on the biomass burning and fossil fuel sources considering that sulphate and resuspended dust factors
were less significant also in terms of EVF for optical variables, ranging from 0.08 to 0.12 and from 0.03 and 0.06,
respectively, depending on the wavelength.
In Fig. 6 the wavelength dependence of $b_{ap}$ for the biomass burning and the fossil fuel profiles obtained with the multi-
time resolution model is shown; as $\alpha$ values can show significant differences when calculated using different pairs of $\lambda$
(Sandradewi et al., 2008b), here we performed a fitting procedure considering $b_{ap} \propto \lambda^{-\alpha}$. Results were $\alpha_{BB}$ ($\alpha$ biomass
burning) = 1.83 and $\alpha_{FF}$ ($\alpha$ fossil fuels) = 0.80; the range of variability of $\alpha$ values was estimated with the bootstrap
analysis obtaining 0.78-0.88 for $\alpha_{FF}$ and 1.65-1.88 for $\alpha_{BB}$ (as 25th and 75th percentile, respectively).

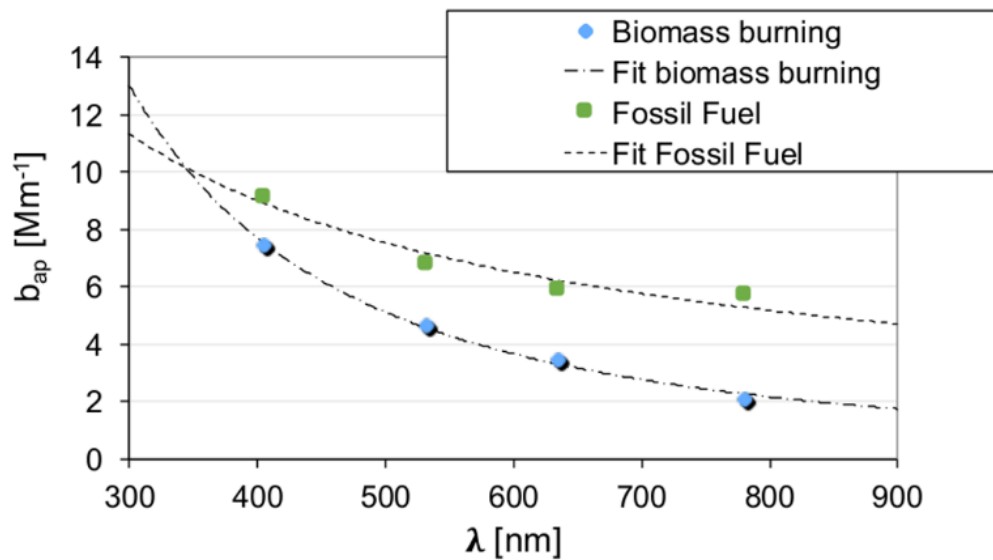


Figure 6: $b_{ap}$ dependence on $\lambda$ for biomass burning and fossil fuel emissions.

Zotter et al. (2017) reported a possible combination of $\alpha_{FF}$=0.8 and $\alpha_{BB}$=1.8 when EC concentration from fossil fuel
combustion (estimated with radiocarbon measurements) is between 40 % and 85 % of the total EC concentration; in this
work, the fraction of EC ascribed by the multi-time model to fossil fuel sources was 56 %. In the same work, the
assessment of $\alpha_{BC}$ (assumed to be equal to $\alpha_{FF}$ in source apportionment optical models) is still an issue and both
experimental and simulation studies are in progress to reduce uncertainties and give a better evaluation of this relevant
optical parameter.
The $\alpha_{BB}$ value retrieved by the model was very similar to values reported by Zotter et al. (2017) and also comparable to
1.86 found for biomass burning by Sandradewi et al. (2008a) and 1.8 obtained by Massabò et al. (2015) who used also
independent $^{14}$C measurements for checking. The $\alpha_{FF}$ value resulted in the range 0.8-1.1 typically reported in optical
source apportionment studies (e.g. Bernardoni et al., 2017b; Zotter et al., 2017; and references therein). Indeed, the
sampling site was an urban background station in Milan where aerosol aging is a relevant process and our samples hardly
had been impacted by fresh traffic emissions. Considering this feature of Milan aerosol, the average $\alpha_{FF}$ was included in
the wide range of estimates for BC coated particles reported in literature works (approx. 0.6-1.3, see e.g. Liu et al., 2018)
and obtained by both ambient measurement (e.g. Fischer and Smith, 2018; and references therein) and numerical
simulations (e.g. Gyawali et al., 2009; Liu et al. 2018; and references therein).
Results here reported allow also to study the relationship between the absorption coefficient and the mass of black carbon,
i.e. the so called Mass Absorption Cross section (MAC) at different wavelengths. The MAC($\lambda$) =$b_{ap}$($\lambda$)/BC relationship
assumes that black carbon (BC) is the only light-absorbing species present; however, this assumption is not always valid,
since mineral dust and brown carbon (BrC) can significantly contribute to aerosol absorption. During our monitoring
campaign, no significant contribution from mineral dust was observed; opposite, biomass burning was proved to be a
relevant source so that BrC was certainly a significant contributor (Fuzzi et al., 2015) as also suggested by $\alpha_{BB}$ = 1.83 in
the biomass burning factor. The possible overestimation of BC when total $b_{ap}$ is ascribed to BC only is usually minimised
choosing a wavelength higher than 600 nm, exploiting the spectral dependence of absorption from different aerosol
compounds (Petzold et al., 2013).
EC concentration retrieved from the chemical profiles (see Fig. 3) was used as a proxy for BC to estimate source-
dependent $b_{ap}$($\lambda$)-to-BC ratio. Results are represented in Fig. 7. It is noteworthy that here this ratio is intentionally not
indicated as MAC, since overestimation of the BC absorption especially at lower $\lambda$ might occur (see previous discussion).
BrC is expected to give a small contribution in the fossil fuel source; therefore, the best approximation for MAC($\lambda$) values
are likely the $b_{ap}(\lambda)$-to-BC ratios observed in the fossil fuel source at our monitoring site. They resulted to be 13.7 m$^2$ g$^{-1}$
at $\lambda$ = 405 nm; 10.2 m$^2$ g$^{-1}$ at $\lambda$ = 532 nm; 8.8 m$^2$ g$^{-1}$ at $\lambda$ = 635 nm; 8.6 m$^2$ g$^{-1}$ at $\lambda$ = 780 nm. At $\lambda$ = 550 nm Bond and
Bergstrom (2006) report MAC = 7.5 $\pm$ 1.2 m$^2$ g$^{-1}$ for uncoated fresh emitted particles and MAC values in polluted regions
ranging from 9 to 12 m$^2$ g$^{-1}$, attributable to absorption enhancement due to particles coating. The MAC estimate obtained
in this work from multi-time resolution model at 532 nm is comparable to literature values and it confirms the importance
of aging processes in atmosphere on the optical properties of particles.

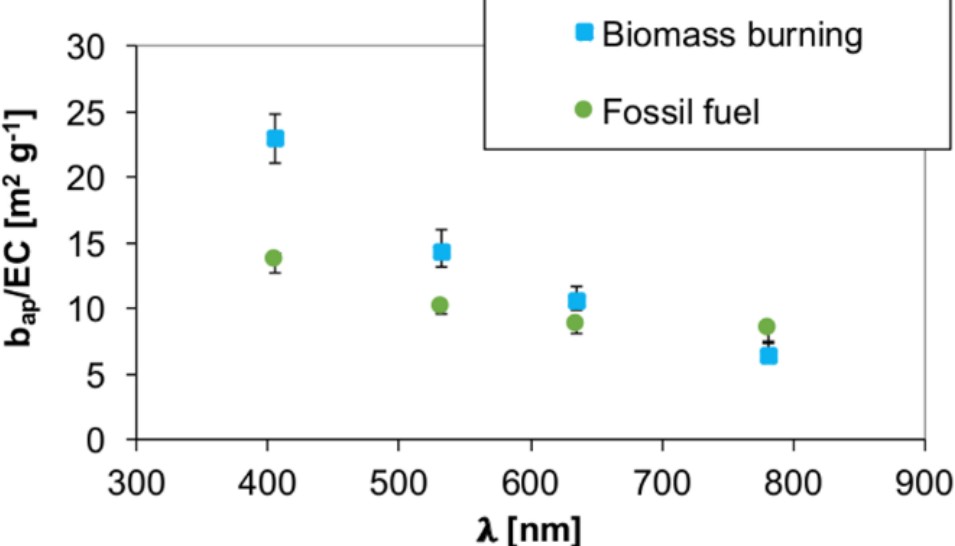


Figure 7: $b_{ap}$-to-EC ratio dependence on $\lambda$ for biomass burning and fossil fuel emissions. Error bars represent the 25[th] and
75[th] percentile retrieved from the bootstrap analysis.

Ratios in Fig. 7 are less comparable at $\lambda$=405 nm (see also Table S4, in the Supplement) due to the significant contribution
of BrC to $b_{ap}$ at this wavelength in the biomass burning factor.
No seasonal differences in the atmospheric ratios were observed but at $\lambda$ = 405 nm (see Table S4, in the Supplement), for
which winter values are higher than summer ones (17.8 $\pm$ 0.4 and 14.2 $\pm$ 0.5, respectively); this result can be explained
considering the influence of biomass burning emissions on BrC concentration in atmosphere during the winter season.
From the outputs of the modelling approach here proposed, the apportionment of the biomass burning and fossil fuel
contributions to $b_{ap}$ at different wavelengths was also obtained. As expected, the relative contribution to the total
reconstructed $b_{ap}$ ascribed to the biomass burning factor decreases with increasing $\lambda$, opposite to the contribution from
fossil fuel combustion which gives the highest contribution at 780 nm (Table 2); in addition, the latter contribution prevails
at all wavelengths at the investigated site.

| | $\lambda = 405$ nm | $\lambda = 532$ nm | $\lambda = 635$ nm | $\lambda = 780$ nm |
|---|---|---|---|---|
| **Biomass burning** | 36 % (31 %-36 %) | 29 % (25 %-30 %) | 26 % (23 %-27 %) | 20 % (16 %-22 %) |
| **Fossil fuels** | 45 % (41 %-46 %) | 43 % (39 %-44 %) | 45 % (41 %-47 %) | 55 % (48 %-55 %) |

Table 2: Average contribution to total reconstructed $b_{ap}$ for the biomass burning and fossil fuel factors; in parenthesis 25[th] and 75[th] percentile are reported.


**4. Conclusions**
The multi-time resolution model implemented through Multilinear Engine (ME2) script allowed the analysis of
experimental data collected at different time scales, coupling the detailed chemical speciation at low time resolution and
the temporal information given by high time resolution samples. The effect of the introduction of the aerosol absorption
coefficient ($b_{ap}$) measured at different wavelengths in the modelling process was investigated and gave promising results.
First of all, a more robust identification of sources was provided; secondly, it paved the way to the retrieval of optical
apportionment and optical characterisation of the sources (e.g. estimate of source-specific Absorption Ångström Exponent
- $\alpha$ - and MAC at different wavelengths). It is worthy to note that – at the state of the art – in source apportionment optical
models (e.g. Aethalometer model) values for $\alpha$ related to fossil fuel emissions and biomass burning are fixed by the
modeller thus carrying a large part of the uncertainties in the model results. Considering that the estimates for the
Absorption Ångström Exponent were here obtained as a result of a quite complex modelling approach (i.e. using multi-
time resolution datasets collected on limited periods) and without any a-priori assumption, the results obtained – although
obviously affected by a certain degree of uncertainty due to both experimental data and modelling process (here estimated
while typically not taken into consideration for fixed $\alpha$ values used in the literature) – were fairly comparable to literature
results and gave a further tool aimed at assessing more robust source-related $\alpha$ values. In perspective, joining together
different approaches such as the receptor modelling here proposed and e.g. [14]C measurements and artefact-free $b_{ap}$
measurements will lead to better estimates of the Absorption Ångström Exponent; work is in progress at our laboratories
to achieve this goal.
The original approach described in this work can be applied to any source apportionment study using any suitable dataset
(not necessarily with multi-time resolution). Besides the traditional source apportionment, the impact of different sources
on the aerosol absorption coefficient was estimated; this piece of information can be very useful to formulate strategies
of pollutants abatement, in order to improve air quality and to face climate challenges. In particular, at the investigated
site secondary compounds constituted the highest contribution in terms of PM10 mass (52 % on average), while the two
factors identified as biomass burning and traffic were found to be the most significant contributors to aerosol absorption
in atmosphere, in agreement with available literature works.

**Acknowledgements**
This work was partially funded by the Italian National Institute of Nuclear Physics under the INFN experiments
DEPOTMASS and TRACCIA. ACTRIS-IT funded the publication of the paper. The authors thank Prof. Paola Fermo
(Dept. of Chemistry, University of Milan) for availability of the Sunset instrument to perform EC/OC analyses and ARPA
– Lombardia for meteorological data availability. The mechanical workshop of the Dept. of Physics – University of Milan
is gratefully acknowledged for the realisation of parts of the polar photometer. The authors are grateful to Prof. Philip
Hopke for hints on multi-time resolution ME-2.

**Data availability.**
The data in the study are available from the authors upon request (roberta.vecchi@unimi.it).

**Supplement.**
The supplement related to this article is available online

**Author contributions.**
ACF performed streaker sampling and related optical analysis, implemented the advanced model, analysed the results,
and drafted the paper. GV contributed to model implementation, data reduction and Hysplit back-trajectories retrieval.
VB, SV, and REP carried out the sampling campaign on filters, performed the optical measurements and data analysis.
GC, SN, and FL performed PIXE analysis and data reduction. DM and PP carried out ionic characterisation on filters and
data analysis. RV was responsible for the design and coordination of the study, the synthesis of the results and the final
version of the paper. All authors contributed to the interpretation of the results obtained with the new approach here
described and revised the manuscript content giving a final approval of the version to be submitted. RV and ACF reviewed
the paper addressing reviewers' comments.

**Competing interests.**
The authors declare that they have no conflict of interest.

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

**List of Captions**
Figure 1: Diurnal profile of Fe and Cu concentrations (in ng m$^{-3}$).
Figure 2: Diurnal profile of the aerosol absorption coefficient measured at different wavelengths in Mm$^{-1}$).
Figure 3: (a) Chemical profiles of the 8-factor constrained solution (b) $b_{ap}$ apportionment of the 8-factor constrained
solution.
Figure 4: Temporal patterns of aged sea salt source retrieved from the multi-time resolution model and Cl concentrations
measured in atmospheric aerosol.
Figure 5: Box plot of the bootstrap analysis on the 8-factor constrained solution. The red dots represent the output values
of the solution of the model; the black lines the medians from the bootstrap analysis; the blue bars the 25$^{th}$ and 75$^{th}$
percentile; the dotted lines the interval equal to 1.5 the interquartile range and the black dots the outliers from this interval.
Figure 6: $b_{ap}$ dependence on $\lambda$ for biomass burning and fossil fuel emissions.
Figure 7: $b_{ap}$-to-EC ratio dependence on $\lambda$ for biomass burning and fossil fuel emissions. Error bars represent the 25$^{th}$ and
75$^{th}$ percentile retrieved from the bootstrap analysis.

Table 1: Absolute and relative average source contributions to PM10 mass in the 8-factor constrained solution.
Table 2: Average contribution to total reconstructed $b_{ap}$ for the biomass burning and fossil fuel factors; in parenthesis 25$^{th}$
and 75$^{th}$ percentile are reported.