# Peer review of "Exploiting multi-wavelength aerosol absorption coefficients in a"

_Atmospheric Chemistry and Physics, 2019_

## Referee Comment (RC1) · Anonymous Referee #1 · 8 Apr 2019

This is a paper which succeeds in combining two areas of investigation in a rather successful way. It takes multi-component chemical data from atmospheric aerosol collected in Milan and processes it together with multi-wavelength optical absorption data in a single analysis using the multi-linear engine ME-2. As well as successfully combining two different kinds of data with different metrics, which has been done before but not for these specific metrics (more usually for particle mass and number size distribution data), it also successfully combines data measured over different averaging periods. The latter is not entirely novel but there are only a small number of earlier

reports in the literature. Consequently, this is advanced receptor modelling work which shows both how using the multi-wavelength optical absorption data from an aethalometer can strengthen source apportionment of light-absorbing components and also that the data can be used in reverse to estimate the optical properties of particles from specific sources.

The paper is in general well written although some aspects of the English could be improved. My main criticism of the science is the lack of detail over the methods. For example, the procedures appear to be successful in combining elemental data and optical absorption data with entirely different metrics and yet outputting concentrations and explained variation for both types of constituent in their original units. This has previously posed problems for PMF but presumably also for ME-2. Secondly, there is no information on whether an error matrix was constructed, and if so, how this was carried out. There are fairly widely accepted methods for chemical data, but how was this achieved for the optical absorption data?

The assignment of identity to the eight factors output by the ME-2 looks very reasonable but there are some specific points that are not addressed. The sulphate factor contains a higher concentration of organic carbon than of sulphur and possibly a higher concentration of organic matter than sulphate (although this is not possible to read from the graph). No mention is made of this organic matter which accounts for a significant proportion of the explained variation of the optical absorption. Presumably this is secondary organic matter correlated with sulphate, but does it have light absorbing properties which are of interest? As various workers have pointed out, this creates problems for the two component "aethalometer" model widely used for source apportionment of wood smoke (but which is not in itself a problem for the ME-2 method). The resuspended dust also appears to have some optical absorption. Could this be associated with the iron minerals? The aged sea salt factor contains substantially more sulphate than sodium and unusually absolutely no nitrate. Although it is shown that the temporal variation of this factor correlates with that of chloride, it appears that this

factor is very atypical of aged sea salt and may well be mixed with other components.

The derivation of Ångström coefficients from the apportioned optical absorption data is interesting but there is little comment on the fact that the value for fossil fuel is 0.78 to 0.88 (25th-75th percentile) which extends slightly below the range of typically reported values and is distinctly different from the value of 1.0 used by most workers in the "aethalometer" model. The value of Ångström coefficients for the biomass burning factor is well within the very wide range of literature values which depend very much upon combustion conditions and is a useful addition to the literature, as are the estimated mass absorption cross-sections.

In addition to addressing the points above, there are two lesser issues which should be considered.

(a) Line 254 – the intercept requires units to be meaningful.

(b) Line 307 – it is stated that "in the factor interpreted as nitrate the explained variation is fully ascribed to NO3-". Would it not be more correct to state that the nitrate factor accounts for 100% of the explained variation in NO3-?

---

## Referee Comment (RC2) · Anonymous Referee #3 · 6 Jun 2019

The article describes a new method to determine source-dependent absorption parameters using a receptor model based on multiple time resolution data (Crespi et al., 2016). Determination of source-dependent Angstrom exponents using C14 technique (Zotter et al., 2017) is time consuming and expensive. There is a need for alternative methods, such as the one proposed by Crespi et al. (2016).

The receptor model was run using data from the offline analysis of chemical composition and the absorption coefficient of particles collected on filters. The absorption was measured using a custom polar photometer which was validated using MAAP at

a single wavelength (Vechi et al., 2014; Bernardoni et al., 2017c). Measurements at 4 different wavelengths are used to calculate the absorption Angstrom exponent. It is not clear how accurate is the value of Angstrom exponent measured by polar photometer as it has not been compared to other multi-wavelength methods, for example: photo-acoustic spectroscopy (Lack et al., 2006), extinction minus scattering, filter photometry (Moosmuller et al., 2009) or being calibrated with laboratory generated aerosol samples (carbon black, nigrosin ...) with known spectral absorption properties and size distribution.

The article proposes a new set of values for the source specific Angstrom exponents for biomass burning and fossil fuel combustion: 1.83 and 0.8, respectively. The biomass burning value is in agreement with previous studies (Zotter et al., 2017), but the value for fossil is quite low. These results are not un-expected as the summer campaign resulted in the average absorption Angstrom exponent as low as 0.58 (data from Table S2) – this value lies outside the range measured in a different settings in Europe. It is also strange that there is no wavelength dependency in absorption coefficients between 532 and 780 nm for the traffic component (Figure 1). Authors should provide an extensive quality control assessment of absorption data to support their findings.

Raw measurements (time-series, diurnal profile) should be presented and discussed before proceeding to the data evaluation using the receptor model.

The language in certain parts of the article is not clear. Here are some specific remarks:

- the parameter describing the spectral dependence of the absorption coefficient is usually named Absorption Angstrom Exponent (similar to Scattering Angstrom Exponent).

- the term Multi-time source apportionment study is not clear. The term Multi-time resolution source apportionment is proposed.

- Line 20: MAC should be defined as Mass Absorption Cross-section instead of Mass Absorption Coefficient

I recommend performing another review after extensive revision of the manuscript. Concerning the scope of the journal, the article might be better suited for the publication in AMT.

---

## Referee Comment (RC3) · Anonymous Referee #2 · 14 Jun 2019

This article presents a source apportionment analysis within the urban area of Milan, Italy. The originality of this study firstly lies in the use of data with different temporal resolutions as one single input data. A modified equation of the standard equation of the MultilinearEngine has been used, which has been scarcely used in the literature. The second original feature is the use of absorption data. The authors were therefore able to derive optical properties of the obtained factors.

Overall, the paper is well-written and is well-organized. I think it deserves publication in ACP, but several points need to adressed before. I fully agree with reviewer #1 about

the methods section. The authors need to be much more didactic on the way they handled uncertainties: - First, little information are provided regarding the calculation of uncertainties for each variable. Sometimes a range of values is provided, but we don't know which value was actually used with the Polissar equation. - Also, since absorption data are rarely used in PMF analyses, I would recommend to perform sensitivity tests on the uncertainty of 15% that was used, and evaluate the impact on the PMF results. - Then, the reader has no information about the balance of the Q in the input variables, yet being a critical issue in the multi-time algorithm. Have the authors adjusted the uncertainties so that the Q is approximatly balanced in each group of variables ? - The authors state that scaled residuals are randomly distributed between -3 and 3. Are these residuals centered around 0, with a Gaussian shape ?

I am also a bit disappointed to see that discussions about optical properties are essentially focused on traffic and wood-burning, but little is said about absorption found in the nitrate-rich factor, the sulfate-rich factor (the presence of EC in the profile is not discussed) and the dust factor.

Finally, in order to strengthen the interpretation of the factors, I recommend to perform a trajectory analysis (eg Potential Source Contribution Function, or Concentration-Weighted Trajectory), especially for aged sea salt and dust. The approach proposed in the manuscript is a bit simplistic.

Specific comments: - p12 l325 : why road dust does thus not appear in the traffic factor ? - p16 l386 : in this paragraph, I would also mention PMF studies including "Delta-C" (Wang et al., 2012), as written in the introduction.

---

## Author Comment (AC1) · 18 Jul 2019

**MS No.: acp-2019-123**

**Title:** *Exploiting multi-wavelength aerosol absorption coefficients in a multi-time resolution source apportionment study to retrieve source-dependent absorption parameters*
Authors: Alice C. Forello et al.

**Response to Reviewers**

The authors acknowledge reviewers for their comments and suggestions, which helped the authors in improving the paper. The authors changed text and figures according to the concerns raised by all referees. English language and grammar was thoroughly revised.

**RESPONSE TO REVIEWER 1**

**R1-1.** This is a paper which succeeds in combining two areas of investigation in a rather successful way. It takes multi-component chemical data from atmospheric aerosol collected in Milan and processes it together with multi-wavelength optical absorption data in a single analysis using the multi-linear engine ME-2. As well as successfully combining two different kinds of data with different metrics, which has been done before but not for these specific metrics (more usually for particle mass and number size distribution data), it also successfully combines data measured over different averaging periods. The latter is not entirely novel but there are only a small number of earlier reports in the literature. Consequently, this is advanced receptor modelling work which shows both how using the multi-wavelength optical absorption data from an aethalometer can strengthen source apportionment of light-absorbing components and also that the data can be used in reverse to estimate the optical properties of particles from specific sources.

**A1-1.** *It is worth noting that the approach here presented is of general interest as (1) in this work optical data were retrieved by a home-made multi-wavelength polar photometer but – as underlined by the Referee - the methodology here presented could be applied to datasets combining aerosol chemical and optical data obtained by widespread instrumentation (e.g. Aethalometers for optical data); (2) input data to the receptor model not necessarily should comprise variables acquired with different time resolution as we did here. We added a sentence on the general character of our work in the revised version of the paper (see lines 38-40 in the abstract and 106-110 afterwards).*

**R1-2.** The paper is in general well written although some aspects of the English could be improved. My main criticism of the science is the lack of detail over the methods. For example, the procedures appear to be successful in combining elemental data and optical absorption data with entirely different metrics and yet outputting concentrations and explained variation for both types of constituent in their original units. This has previously posed problems for PMF but presumably also for ME-2.

**A1-2.** *ME-2/PMF analysis is not a-priori harmed by the use of joint matrices containing different units, as we ascertained after having studied a number of published papers/users' guides before joining optical and chemical data in the same input matrix. Recently, this point has been also extensively discussed and explained by P. Paatero in an open discussion on ACPD (see https://doi.org/10.5194/acp-2018-784-RC2). Indeed, as he stated, if different units are present in different columns of matrix X, the output data in factor matrix G are pure numbers and elements in a column of factor matrix F carry the same dimension and unit as the original data in matrix X. In*

*addition, as we did in this work, the average total contribution to the mass of a specific source due to species in a certain factor in matrix F must be retrieved a-posteriori summing up only mass contributions by chemical components (i.e. excluding optical components in matrix F).*

*Following the Referee's comment, we decided to add a short sentence about this point which is often considered an issue in PMF/ME analysis but it is mainly due to a misunderstanding (see lines 302-307 in the revised version).*

**R1-3.** Secondly, there is no information on whether an error matrix was constructed, and if so, how this was carried out. There are fairly widely accepted methods for chemical data, but how was this achieved for the optical absorption data?

**A1-3.** *In the revised version of the paper, we added experimental uncertainties and MDLs (as reported for each analysis in sections 2.2 and 2.3) and some sentences hopefully clarify how uncertainties were handled in the model. See e.g.*

*Lines 138-140: In this Section, chemical analyses performed on samples are summarised. As measured concentration in each sample was characterised by its own uncertainty, only ranges for experimental uncertainties and minimum detection limits (MDLs) for every set of variables are reported.*

*Lines 168-171: PM10 hourly concentrations of most elements and samples were characterised by relative uncertainties in the range 10-30% (higher uncertainties for elements near MDL) and MDLs ranged from a minimum of 0.1 to a maximum of 2.5 ng m$^{-3}$ (higher MDLs typically detected for Z<20 elements).*

*Lines 198-201: Uncertainties on $b_{ap}$ were estimated as 15 % and MDL was in the range 1-10 Mm$^{-1}$ depending on sampling duration and wavelength as already reported in our previous works (Vecchi et al., 2014; Bernardoni et al., 2017c). Experimental uncertainties and MDL of optical absorption data were used as a starting point to estimate the uncertainties introduced in the model. Pre-treatment procedure for these data was the same used for chemical variables (see also Sect. 2.5).*

*Lines 269-273: Every measured variable in each sample is characterised by its own uncertainty; ranges of experimental uncertainties and MDLs are reported in Sect. 2.2 and 2.3 for chemical and optical analyses, respectively. Variables with more than 20 % of the concentration data below MDL values were omitted from the analysis (Ogulei et al., 2005). The procedure described in Polissar et al. (1998) was followed to treat uncertainties and below MDL data, starting from experimental uncertainties and MDLs.*

*Lines 288-291: The input matrix X consisted in 386 samples and the total number of time units was 1117. The analysis was performed in the robust mode; lower limit for G contribution was set to -0.2 (Brown et al., 2015) and the error model em=-14 was used for the main equation with C1= input error, C2= 0.0 and C3=0.1 (Paatero, 2012) for both chemical and optical absorption data.*

**R1-4.** The assignment of identity to the eight factors output by the ME-2 looks very reasonable but there are some specific points that are not addressed. The sulphate factor contains a higher concentration of organic carbon than of sulphur and possibly a higher concentration of organic matter than sulphate (although this is not possible to read from the graph).

**A1-4.** *The concern raised by the Referee was addressed discussing a bit further this point in the revised text (see lines 391-404): "The factor associated to sulphate shows EVF=0.47 for S and much lower EVF for all the other variables in the factor. Considering the contribution of S in the chemical profile in terms of sulphate and ammonium sulphate, the relative contribution of sulphur components*

*in the profile increases from 11% (S) up to 45% (ammonium sulphate). The latter is the main sulphur compound detected in the Po valley as reported in previous papers such as e.g. Marcazzan et al. (2001) and was by far the highest contributor in the chemical profile. The other important contributor was OC (19%), whose impact on PM mass increased up 30% when reported as organic matter using 1.6 as the organic carbon-to-organic matter conversion factor for this site (Vecchi et al., 2004). Due to the secondary origin of the aerosol associated to this factor, it was not surprising to find also a significant OC contribution; indeed, aerosol chemical composition in Milan is impacted by highly oxygenated components due to aging processes favoured by strong atmospheric stability (Vecchi et al., 2018 and 2019). In this factor, EC contributed for about 1%. Considering the total EC concentration reconstructed by the model, the EC fraction related to the sulphate factor was about 6%. Opposite to sulphates, EC has a primary origin; however, its presence with a very similar percentage (4-5%) in a sulphate chemical profile was previously pointed out in Milan, indicating a more complex mixing between primary and secondary sources (Amato et al., 2016). The sulphate factor accounted for 21 % of the PM10 mass.''*

**R1-5.** No mention is made of this organic matter which accounts for a significant proportion of the explained variation of the optical absorption. Presumably this is secondary organic matter correlated with sulphate, but does it have light absorbing properties which are of interest? As various workers have pointed out, this creates problems for the two component "aethalometer" model widely used for source apportionment of wood smoke (but which is not in itself a problem for the ME-2 method). The resuspended dust also appears to have some optical absorption. Could this be associated with the iron minerals?

**A1-5.** *To address the Referee's concern, in the revised text we discussed a bit more results related to the other contributors to aerosol absorption in atmosphere although their contribution is not significant in terms of EVF (ranging from 0.08 to 0.12 for sulphate and from 0.03 to 0.06 for resuspended dust, depending on the wavelength).*

*Lines 497-508: "The third contributor to aerosol absorption in atmosphere was the sulphate factor, with a contribution comparable to the biomass burning one at 780 nm (about 20% of the total reconstructed bap at this wavelength). The sulphate factor contained a small fraction of EC, as previously discussed (see Sect. 3.2). This might be explained considering that non/weakly light-absorbing material can form a coating able to enhance absorption (Bond & Bergstrom, 2006; Fuller et al., 1999) within a few days after emission (Bond et al., 2006). Laboratory experiments and simulations from in-situ measurements highlighted absorption amplification for absorbing particles coated with secondary organic aerosol (Schnaiter et al., 2003; Moffet & Prather, 2009). These processes related to particles aging can become important in the Po valley due to low atmospheric dispersion conditions and they might explain the relatively high contribution of the sulphate factor to the absorption coefficient in respect to the other sources (excluding traffic and biomass burning). Among the other sources, resuspended dust was the main contributor at all wavelengths (between 3% and 7% of the total reconstructed bap, depending on the wavelength), likely due to the role of iron minerals. The other four sources were less relevant in terms of EVF values and overall contributed for less than 11%."*

**R1-6.** The aged sea salt factor contains substantially more sulphate than sodium and unusually absolutely no nitrate. Although it is shown that the temporal variation of this factor correlates with

that of chloride, it appears that this factor is very atypical of aged sea salt and may well be mixed with other components.

**A1-6.** *The Referee is right, being Milan about 120 km far from the nearest coast, when occasionally observed, transported marine air masses are affected by the pollution encountered in the Po valley. This is why the chemical profile of this aged sea salt is dirty and clearly mixed with other components. Nevertheless, this feature is expected and can be explained considering that during air mass transport from the sea to continental polluted areas sodium chloride can react with sulfuric acid vapor, producing sodium sulphate and hydrochloric acid vapor and giving chloride depletion (Seinfeld and Pandis, 2006). As the Referee points out, chloride deficit can also be caused by NaCl reacting with HNO₃ and producing NaNO₃ in particulate phase (Seinfeld and Pandis, 2006). The presence of nitrate and sulphate in aged sea salt chemical profile has been previously reported for a number of sites e.g. by Amato et al. (2016), including also Milan.*

*The aged sea salt source is enriched in sulphate even though its share in this factor is low compared to its overall concentration in PM (10% of the total reconstructed variable). As for the lack of nitrate, following Lee et al. (1999) a very rough estimate of the $NO_3^-$ content that can be ascribed to this factor is from 0% to a maximum of 2% (about 82 ng m⁻³) of the total reconstructed concentration of $NO_3^-$ in atmosphere, that can be considered totally negligible compared to changes in the mass contribution of sources of $NO_3^-$. From the experimental data, it is not possible to obtain information about nitrate transport during marine mass advection: atmospheric concentrations of $NO_3^-$ over 12 or 24 hours comprising the marine advection are always very low, because of "cleaner" air mass (255 ng m⁻³ in summer and 1.4 µg m⁻³ in winter, to be compared with average values during the two seasons of 1.4 µg m⁻³ and 9.0 µg m⁻³, respectively). It is possible that $NO_3^-$ measurements available only at low-time resolution influence its estimation in the aged sea salt episodic source. Anyway, the multi-time model was able to catch the signal and the main tracers of this episodic source.*

*We added a sentence, at lines 447-451: "When marine air masses are transported to polluted sites, sea salt particles are characterised by a Cl deficit due to reactions with sulphuric and nitric acid (Seinfeld and Pandis, 2006). In this case, the factor chemical profile was expected to be enriched in sulphate and nitrate. In this work, nitrate was not present; a very rough estimate (Lee et al., 1999) gave a maximum expected contribution of 2 % (about 82 ng m-3) of the total nitrate mass in atmosphere, that can be considered negligible in terms of mass contribution of the sources."*

**R1-7.** The derivation of Ångström coefficients from the apportioned optical absorption data is interesting but there is little comment on the fact that the value for fossil fuel is 0.78 to 0.88 (25th-75th percentile) which extends slightly below the range of typically reported values and is distinctly different from the value of 1.0 used by most workers in the "aethalometer" model. The value of Ångström coefficients for the biomass burning factor is well within the very wide range of literature values which depend very much upon combustion conditions and is a useful addition to the literature, as are the estimated mass absorption cross-sections.

**A1-7**. *There is an open discussion on the "best" α value to be used in the Aethalometer model as for fossil fuel as in the literature it ranges from 0.8-1.1 (see e.g. Sandradewi et al., 2008 and Zotter et al., 2017). Indeed, $α_{FF}=1$ is derived from the theory of absorption of spherical particles in the Rayleigh regime, but both experimental and modelling studies highlighted the possibility to have also lower values for atmospheric particles due to the influence of particle size distribution and aging processes in atmosphere (as reported at lines 542-545). In addition, in this work aerosol absorption coefficient at different wavelengths is measured on PTFE and polycarbonate filters (i.e. membranes),*

*while reported literature values are usually retrieved from filter-based instrumentation using fibre-filter tapes, which are known to be affected by possible biases. We reported this issue at lines 184-188: "Moreover, previous works reported a bias on $b_{ap}$ measured by instrumentation using fibre filters (e.g. Cappa et al., 2008: Lack et al., 2008; Davies et al., 2019; and references therein); Vecchi et al. (2014) quantified in about 40% the effect caused in $b_{ap}$ values (assessed at 635 nm) by sampling artefacts due to organics in aerosol samples collected in Milan when comparing aerosol samples collected in parallel quartz-fibre and PTFE filters.".*

In addition to addressing the points above, there are two lesser issues which should be considered.

(a)     Line 254 – the intercept requires the units to be meaningful.

*Absolutely right, it has been added.*

(b) Line 307 – it is stated that "in the factor interpreted as nitrate the explained variation is fully ascribed to $NO_3^-$ ". Would it not be more correct to state that the nitrate factor accounts for 100% of the explained variation in NO3-?

*Yes, indeed. It has been corrected in the revised version (line 382) as follows: "The factor interpreted as nitrate fully accounted for the explained variation of $NO_3^-$."*

*References*

- *Amato et al. (2016). AIRUSE-LIFE+: a harmonized PM speciation and source apportionment in five southern European cities. Atmos. Chem. Phys., 16, 3289–3309.*
- *Cappa C.D., Lack D.A., Burkholder J.B., and Ravishankara A.R.: Bias in filter-based aerosol light absorption measurements due to organic aerosol loading: Evidence from laboratory measurements. Aerosol Sci. Tech., 42, 1022-1032.*
- *Davies N.W., Fox C., Szpek K., Cotterell M.I., Taylor J.W., Allan J.D., Williams P.I., Trembath J., Haywood j.M., and Langridge J.M: Evaluating biases in filter-based aerosol absorption measurements using photoacoustic spectroscopy, Aerosol Meas. Tech., 12, 3417–3434.*
- *Lack D.A., Cappa C.D., Covert D.S., Baynard T., Massoli P., Sierau B., Bates T.S., Quinn P.K., Lovejoy E.R., and Ravishankara A.R.: Bias in filter-based aerosol light absorption measurements due to organic aerosol loading: Evidence from ambient measurements. Aerosol Sci. Tech., 42, 1033-1041.*
- *Marcazzan et al. (2001). Characterisation of PM10 and PM2.5 particulate matter in the ambient air of Milan (Italy), Atmos. Environ., 35, 4639-4650.*
- *Paatero P. (2018). Open discussion on ACPD at https://doi.org/10.5194/acp-2018-784-RC2*
- *Sandradewi J., Prévôt A.S.H., Szidat S., Perron N., Alfarra M.R., Lanz V.A., Weingartner E. and Balternsperger U.: Using aerosol light absorption measurements for the quantitative determination of wood burning and traffic emission contributions to particulate matter, Environ. Sci. Technol., 42, 3316-3323, 2008a.*
- *Seinfeld J.H. and Pandis S.N. (2006): Atmospheric chemistry and physics: from air pollution to climate change, 2nd edition, John Wiley & Sons, INC, Hoboken, New Jersey.*
- *Vecchi R., Bernardoni V., Paganelli C. and Valli G.: A filter-based light absorption measurement with polar photometer: effects of sampling artefacts from organic carbon, J. Aerosol. Sci., 70, 15-25, https://doi.org/10.1016/j.jaerosci.2013.12.012, 2014.*
- *Zotter P., Herich H., Gysel M., El-Haddad I., Zhang Y., Mocnik G., Hüglin C., Baltensperger U., Szidat S. and Prévôt A.S.H.: Evaluaton of the absorption Ångström exponents for traffic and wood burning in the Aethalometer-based source apportionment using radiocarbon measurements of ambient aerosol, Atmos. Chem. Phys., 17, 4229-4249, https://doi.org/10.5194/acp-17-4229-2017, 2017.*

**R2-1.** This article presents a source apportionment analysis within the urban area of Milan, Italy. The originality of this study firstly lies in the use of data with different temporal resolutions as one single input data. A modified equation of the standard equation of the Multilinear Engine has been used, which has been scarcely used in the literature. The second original feature is the use of absorption data. The authors were therefore able to derive optical properties of the obtained factors.

Overall, the paper is well-written and is well-organized. I think it deserves publication in ACP, but several points need to addressed before.

I fully agree with reviewer #1 about the methods section. The authors need to be much more didactic on the way they handled uncertainties: - First, little information are provided regarding the calculation of uncertainties for each variable. Sometimes a range of values is provided, but we don't know which value was actually used with the Polissar equation.

**A2-1** *In the revised version of the paper, we added experimental uncertainties and MDL (as reported for each analysis in sections 2.2 and 2.3) and some sentences hopefully clarify how uncertainties were handled in the model. See e.g.*

*Lines 138-140: In this Section, chemical analyses performed on samples are summarised. As measured concentration in each sample was characterised by its own uncertainty, only ranges for experimental uncertainties and minimum detection limits (MDLs) for every set of variables are reported.*

*Lines 168-171: PM10 hourly concentrations of most elements and samples were characterised by relative uncertainties in the range 10-30% (higher uncertainties for elements near MDL) and MDLs ranged from a minimum of 0.1 to a maximum of 2.5 ng m$^{-3}$ (higher MDLs typically detected for Z<20 elements).*

*Lines 198-201: Uncertainties on $b_{ap}$ were estimated as 15 % and MDL was in the range 1-10 Mm$^{-1}$ depending on sampling duration and wavelength as already reported in our previous works (Vecchi et al., 2014; Bernardoni et al., 2017c). Experimental uncertainties and MDL of optical absorption data were used as a starting point to estimate the uncertainties introduced in the model. Pre-treatment procedure for these data was the same used for chemical variables (see also Sect. 2.5).*

*Lines 269-273: Every measured variable in each sample is characterised by its own uncertainty; ranges of experimental uncertainties and MDLs are reported in Sect. 2.2 and 2.3 for chemical and optical analyses, respectively. Variables with more than 20 % of the concentration data below MDL values were omitted from the analysis (Ogulei et al., 2005). The procedure described in Polissar et al. (1998) was followed to treat uncertainties and below MDL data, starting from experimental uncertainties and MDLs.*

*Lines 288-291: The input matrix X consisted in 386 samples and the total number of time units was 1117. The analysis was performed in the robust mode; lower limit for G contribution was set to -0.2 (Brown et al., 2015) and the error model em=-14 was used for the main equation with C1= input error, C2= 0.0 and C3=0.1 (Paatero, 2012) for both chemical and optical absorption data.*

**R2-2.** Also, since absorption data are rarely used in PMF analyses, I would recommend to perform sensitivity tests on the uncertainty of 15% that was used, and evaluate the impact on the PMF results.

**A2-2**. *Sensitivity tests on the uncertainty of absorption data were performed starting from a minimum uncertainty of 10%. Lower uncertainties were considered not physically meaningful from an experimental point of view. ME-2 analyses performed with this minimum uncertainty on absorption*

*data gave very similar results to the base case solution presented in the Supplement (Figure S1 and Table S1), with no differences in mass apportionment and a maximum variation in the concentrations of chemical and optical profiles (matrix F) of 7% considering significant variables in each profile (EVF higher or near 0.30). Opposite, with an uncertainty of 20% on absorption data, the solution corresponding to the minimum Q value was different from the base case one presented in the Supplement. The factors assigned to resuspended dust and construction works got mixed, and a new unique factor (traced almost exclusively by Pb) appeared, with mass contribution equal to zero.*

*A quantitative parameter to estimate how much chemical and optical variables "drive" the model is the "Total weight", recently introduced by P. Paatero in an open discussion on ACPD about joint analyses of matrices of variables with different units (see https://doi.org/10.5194/acp-2018-784-RC2).*

*In our work, 18 chemical variables and 4 optical variables were introduced in the multi-time resolution model (see section 2.5) and the ratio between the total weight of the "chemical variables matrix" and the "optical variables matrix" was about 9, 12, and 19 considering an uncertainty on absorption data of 10%, 15%, and 20%, respectively.*

*In this case, it seems that when the "chemical variables matrix" weights about 19 times more than the optical matrix, it drives the model too much (i.e. relevance of optical variables is weakened and no useful information are added by the joint datasets); opposite, a ratio of 12 is enough to avoid any variable-driven solution. The ratios obtained for a joint analysis of chemical and optical variables clearly depend on the variables used, the matrices dimension and the parameters implemented in the model; more studies are needed to give more robust indication on the "optimum" ratio for this kind of analyses.*

*In conclusion, we considered the experimental uncertainty of 15% as the best option for our optical absorption data. A comment on sensitivity tests results is now reported at lines 292-301.*

**R2-3.** Then, the reader has no information about the balance of the Q in the input variables, yet being a critical issue in the multi-time algorithm. Have the authors adjusted the uncertainties so that the Q is approximatly balanced in each group of variables?

**A2-3**. *Obviously, the number of variables in each time frame will contribute different amounts to the Q. Variables with higher time resolution have more values so they influence Q more than the lower time resolution variables. That is the whole point of the approach. We know that there is more potential for edge points in the high time resolution data since we are not averaging over higher and lower concentrations in the longer time periods. Thus, we want there to be imbalanced in the Q so that the data with the maximum source information drives the solution and those data are the higher time resolution data.*

*To the authors' knowledge there is no literature papers on multi-time resolution ME-2 where different portions of the Q are weighed so that the variables with different time resolutions have equal impact on the solution. That would defeat the purpose of using the data in their native time resolutions.*

*In some models, where there is an auxiliary Q, you want it to have less weight than the main Q, but in the Q itself, you really do not want to start fooling with weighting groups of variables.*

*The authors would like to acknowledge P. Hopke, who shared his expertise with us about this point.*

**R2-4.** The authors state that scaled residuals are randomly distributed between -3 and 3. Are these residuals centered around 0, with a Gaussian shape?

**A2-4**. *The Referee is right, we forgot to comment on the shape of scaled residuals distribution. For each variable, the scaled residuals of the base case solution are now reported in the Supplement (Fig. S2). In addition, at lines 357-359 a sentence was added "with a Gaussian shape for most of the variables (Fig. S2 in the Supplement)."*

**R2-5.** I am also a bit disappointed to see that discussions about optical properties are essentially focused on traffic and wood-burning, but little is said about absorption found in the nitrate-rich factor, the sulfate-rich factor (the presence of EC in the profile is not discussed) and the dust factor.

**A2-5**. *We decided to focus the discussion about optical properties specifically on fossil fuel and biomass burning, in order to be comparable to the approach of the widespread Aethalometer model, as mentioned at lines 519-522.*

*To address the Referee's concern, in the revised text we discussed a bit more results related to the other contributors to aerosol absorption in atmosphere although their contribution is not significant in terms of EVF (ranging from 0.08 to 0.12 for sulphate and from 0.03 to 0.06 for resuspended dust, depending on the wavelength).*

*Lines 497-508: "The third contributor to aerosol absorption in atmosphere was the sulphate factor, with a contribution comparable to the biomass burning one at 780 nm (about 20% of the total reconstructed $b_{ap}$ at this wavelength). The sulphate factor contained a small fraction of EC, as previously discussed (see Sect. 3.2). This might be explained considering that non/weakly light-absorbing material can form a coating able to enhance absorption (Bond & Bergstrom, 2006; Fuller et al., 1999) within a few days after emission (Bond et al., 2006). Laboratory experiments and simulations from in-situ measurements highlighted absorption amplification for absorbing particles coated with secondary organic aerosol (Schnaiter et al., 2003; Moffet & Prather, 2009). These processes related to particles aging can become important in the Po valley due to low atmospheric dispersion conditions and they might explain the relatively high contribution of the sulphate factor to the absorption coefficient in respect to the other sources (excluding traffic and biomass burning). Among the other sources, resuspended dust was the main contributor at all wavelengths (between 3% and 7% of the total reconstructed $b_{ap}$, depending on the wavelength), likely due to the role of iron minerals. The other four sources were less relevant in terms of EVF values and overall contributed for less than 11%."*

*About the presence of EC in the sulphate factor, we added a little discussion in the paragraph regarding this factor.*

*See lines 400-403: "In this factor, EC contributed for about 1%. Considering the total EC concentration reconstructed by the model, the EC fraction related to the sulphate factor was about 6%. Opposite to sulphates, EC has a primary origin; however, its presence with a very similar percentage (4-5%) in a sulphate chemical profile was previously pointed out in Milan, indicating a more complex mixing between primary and secondary sources (Amato et al., 2016)."*

**R2-6.** Finally, in order to strengthen the interpretation of the factors, I recommend to perform a trajectory analysis (eg Potential Source Contribution Function, or Concentration-Weighted Trajectory), especially for aged sea salt and dust. The approach proposed in the manuscript is a bit simplistic.

**A2-6**. *We thank the Referee for suggestion but application of PSCF/CWT needs a previous experience that we have not yet, so that we preferred not to perform an analysis before having a deep knowledge of the tool we are using. Moreover, as far as we have understood, PSCF applied on the whole*

*campaign results hardly could give a realistic picture of marine air masses arriving episodically in Milan (a few events during the year originated from very different marine areas). Maybe that figure S3 (in the previous version of the paper) was a bit confusing so that we changed it with a representation where back-trajectories carrying also information on sea salt concentrations are given focusing on the episode discussed in the text.*

*Moreover, in many source apportionment studies factor interpretation based on chemical tracers, diagnostic ratios, seasonal contributions, etc. is often considered appropriate; in addition, in this study for a couple of factors optical variables and/or additional information not included in the model (e.g. discussion with Cl in the fine and coarse fraction for aged sea salt) strengthen the factor assignment.*

Specific comments:
- p12 l325 : why road dust does thus not appear in the traffic factor ?

*As reported at line 407, the road dust contribution appears to be mixed in the resuspended dust source. At lines 423-425 we added a sentence to underline this information "The lack of relevant crustal elements such as Ca and Al in the chemical profile, suggested a negligible impact of road dust in this factor. As reported above, at our sampling site the road dust contribution was very likely mixed to resuspended dust and further separation of these contributions was not possible."*

- p16 l386 : in this paragraph, I would also mention PMF studies including "Delta-C" (Wang et al., 2012), as written in the introduction.

*We added the following sentence (lines 514-516): "It has to be mentioned that optical models are typically based on a two-source hypothesis (i.e. biomass burning and fossil fuel emissions); an exception reported in previous works (Wang et al., 2011) relies on the use of Delta-C used as an input variable together with chemical aerosol components in source apportionment models and proved to be very effective in separating traffic (especially diesel) emissions from wood combustion emissions".*

**R3-1.** The article describes a new method to determine source-dependent absorption parameters using a receptor model based on multiple time resolution data (Crespi et al., 2016). Determination of source-dependent Angstrom exponents using C14 technique (Zotter et al., 2017) is time consuming and expensive. There is a need for alternative methods, such as the one proposed by Crespi et al. (2016).

**A3-1**. *As already mentioned in the response A1-1 to Referee 1, it is worth noting that the approach here presented is of general interest as (1) in this work optical data were retrieved by a home-made multi-wavelength polar photometer but the methodology here presented could be applied to dataset combining aerosol chemical and optical data obtained by widespread instrumentation (e.g. Aethalometers for optical data); (2) input data to the receptor model not necessarily should comprise variables acquired with different time resolution as we did here. We added a sentence on the general character of our work in the revised version of the paper (see lines 38-40 in the abstract and 106-110 afterwards).*

**R3-2.** The receptor model was run using data from the offline analysis of chemical composition and the absorption coefficient of particles collected on filters. The absorption was measured using a custom polar photometer which was validated using MAAP at a single wavelength (Vecchi et al., 2014; Bernardoni et al., 2017c). Measurements at 4 different wavelengths are used to calculate the absorption Angstrom exponent. It is not clear how accurate is the value of Angstrom exponent measured by polar photometer as it has not been compared to other multi-wavelength methods, for example: photo-acoustic spectroscopy (Lack et al., 2006), extinction minus scattering, filter photometry (Moosmuller et al., 2009) or being calibrated with laboratory generated aerosol samples (carbon black, nigrosin ...) with known spectral absorption properties and size distribution.

**A3-2.** *At the time of the paper submission, a laboratory experiment was in progress to intercompare PP_UniMI with optical instruments other than MAAP. The laboratory experiment was carried out at the Jülich Forschungszentrum (Germany). Preliminary data are currently available but at the moment they have been only presented in a poster by Valentini et al. at the International Conference on Carbonaceous Particles in Atmosphere in Vienna (ICCPA2019, https://iccpa2019.univie.ac.at/abstracts/; a copy of the poster can be requested to the authors). Briefly, samples of laboratory-generated aerosol were collected on filters and measured in parallel by on-line instrumentation. Light extinction at 450 and 630 nm was obtained by two Cavity Attenuated Phase Shift CAPS PMSSA (Aerodyne Research). An integrating Nephelometer (TSI) measured total and back- scattering coefficients at 450, 550, and 700 nm. Filter samples were analysed off-line by our polar photometer PP_UniMI.*

*As far as preliminary results are concerned: the absorption Ångström exponents retrieved by extinction minus scattering method compared to data obtained by PP_UniMI measurements show a very good agreement (always well within 1 standard deviation) as reported in the mentioned poster. Therefore, in the revised version of the paper (lines 176-182) a sentence was added: "Results on $b_{ap}$ obtained by this custom photometer resulted in very good agreement against multi-angle absorption photometer (MAAP) data at 635 nm (Vecchi et al., 2014; Bernardoni et al., 2017c). More recently, in the frame of a collaboration with the Jülich Forschungszentrum (Germany), the Absorption Ångström Exponents retrieved by extinction minus scattering measurements were compared at two wavelengths (630 nm and 450 nm) with the one obtained by PP_UniMI data for laboratory-generated aerosols. The agreement with Cabot soot was in general very good as for both $b_{ap}$ at two wavelengths and Absorption Ångström Exponent estimates, i.e. comparability within one standard deviation (data not yet published, preliminary results reported in Valentini et al., 2019)."*

**R3-3.** The article proposes a new set of values for the source specific Angstrom exponents for biomass burning and fossil fuel combustion: 1.83 and 0.8, respectively. The biomass burning value is in

agreement with previous studies (Zotter et al., 2017), but the value for fossil is quite low. These results are not un-expected as the summer campaign resulted in the average absorption Angstrom exponent as low as 0.58 (data from Table S2) – this value lies outside the range measured in a different settings in Europe.

**A3-3.** *Zotter et al. (2017) in their paper concluded that "it is recommended to use the best α combination as obtained here ($\alpha_{TR}$ = 0.9 and $\alpha_{WB}$ = 1.68) in future studies when no or only limited additional information like $^{14}C$ measurements are available" but they also stated that "a possible combination of $\alpha_{FF}$=0.8 and $\alpha_{BB}$=1.8 when EC concentration from fossil fuel combustion (estimated with radiocarbon measurements) is between 40% and 85% of the total EC concentration" (in this work, the fraction of EC ascribed by the multi-time model to fossil fuel sources was 56%).*

*As now reported at lines 537-545, "The $\alpha_{BB}$ value retrieved by the model was very similar to values reported by Zotter et al. (2017) and also comparable to 1.86 found for biomass burning by Sandradewi et al. (2008a) and 1.8 obtained by Massabò et al. (2015) who used also independent $^{14}C$ measurements for checking. The $\alpha_{FF}$ value (assumed to be equal to $\alpha_{BC}$ in source apportionment optical models) resulted in the range 0.8-1.1 typically reported in optical source apportionment studies (e.g. Bernardoni et al., 2017b; Zotter et al., 2017; and references therein). Indeed, the sampling site was an urban background station in Milan where aerosol aging is a relevant process and our samples hardly had been impacted by fresh traffic emissions. Considering this feature of Milan aerosol, the average $\alpha_{FF}$ was included in the wide range of estimates for BC coated particles reported in literature works (approx. 0.6-1.3, Liu et al., 2018) and obtained by both ambient measurement (e.g. Fisher and Smith, 2018; and references therein) and numerical simulations (e.g. Gyawali et al., 2009; Liu et al. 2018; and references therein)."*

*Therefore, we concluded that (see lines 589-597): "Considering that the estimates for the Absorption Ångström Exponent were here obtained as a result of a quite complex modelling approach (i.e. using multi-time resolution datasets collected on limited periods) and without any a-priori assumption, the results obtained – although obviously affected by a certain degree of uncertainty due to both experimental data and modelling process (here estimated while often not taken into consideration for fixed α values used in the literature) – were fairly comparable to literature results and gave a further tool aimed at assessing more robust source-related α values. In perspective, joining together different approaches such as the receptor modelling here proposed and e.g. $^{14}C$ measurements and artefact-free $b_{ap}$ measurements will likely lead to better estimates of the Absorption Ångström Exponent; work is in progress at our laboratories to achieve this goal."*

*In literature works, it is rarely taken into account that α is also very sensitive to small variations in $b_{ap}$ values; indeed, considering α estimates based on experimental data with e.g. uncertainties of about 10% (which is a quite common estimate for uncertainty associated to absorption optical measurements) effects on the retrieved value can be relevant.*

**R3-4.** It is also strange that there is no wavelength dependency in absorption coefficients between 532 and 780 nm for the traffic component (Figure 1). Authors should provide an extensive quality control assessment of absorption data to support their findings.

**A3-4.** *: The Referee is right but it must be taken into account that $b_{ap}$ was splitted by the model into different sources with a degree of uncertainty given by the modelling approach itself. This is the first work coupling multi-time resolution and multi-variable datasets so that further applications – e.g. with larger temporal coverage and from different sites – will likely indicate us which is the parameter to be optimized in the model to retrieve the best overall result. However, as already mentioned, the estimates here reported can be surely considered not less robust than those often taken a-priori in other literature works.*

**R3-6.** Raw measurements (time-series, diurnal profile) should be presented and discussed before proceeding to the data evaluation using the receptor model.

**A3-6**.: *Following the Referee's request we added a section (Sect. 3.1 in the revised text) where a general description of the dataset is given. As for time series of raw measurements, we decided to present them only when necessary for sources interpretation (see the aged sea-salt source, Figure 2 and S3) in order to avoid redundant information.*

**R3-7.** The language in certain parts of the article is not clear.
**A3-7.** *English language and grammar was thoroughly revised*

Here are some specific remarks:
- the parameter describing the spectral dependence of the absorption coefficient is usually named Absorption Angstrom Exponent (similar to Scattering Angstrom Exponent).
*Correction done*
- the term Multi-time source apportionment study is not clear. The term Multi-time resolution source apportionment is proposed.
*Following literature papers dealing with this kind of analysis we omitted the term "resolution"; however, for clarity it has been now added.*
- Line 20: MAC should be defined as Mass Absorption Cross-section instead of Mass Absorption Coefficient
*Correction done.*
I recommend performing another review after extensive revision of the manuscript. Concerning the scope of the journal, the article might be better suited for the publication in AMT.
*This kind switch can be done at the initial submission stage and not after the long discussion phase on ACPD. We would like also to remark that both Referees at the initial stage evaluated the paper suitable for publication on ACPD.*

*References*
- *Bernardoni V., Valli G. and Vecchi R.: Set-up of a multi-wavelength polar photometer for the off-line measurement of light absorption properties of atmospheric aerosol collected with high-temporal resolution, J. Aerosol. Sci., 107, 84-93, 2017*
- *Fischer D.A. and Smith G.D.: A portable, four wavelength, single-cell photoacoustic spectrometer for ambient aerosol absorption, Aerosol Sci. Tech., 52, 393-406, 2018*
- *Gyawali M., Arnott W.P., Lewis K. and Moosmüller H.: In situ aerosol optics in Reno, NV, USA during and after the summer 2008 California wildfires and the influence of absorbing and non-absorbing organic coatings on spectral light absorption, Atmos.Chem.Phys, 9, 2009*
- *Liu C., Chung C.E., Yin Y. and Schnaiter M.: The absorption Ångström exponent of black carbon: from numerical aspects. Atmos. Chem. Phys., 18, 6259-6273, 2018*
- *Valentini et al. (2019); Multi-Wavelength Measurement of Aerosol Optical Properties: Laboratory Intercomparison of In-Situ and Filter-Based Techniques; 12th International Conference on Carbonaceous Particles in the Atmosphere (ICCPA) 2019   https://iccpa2019.univie.ac.at/abstracts/*
- *Vecchi R., Bernardoni V., Paganelli C. and Valli G.: A filter-based light absorption measurement with polar photometer: effects of sampling artefacts from organic carbon, J. Aerosol. Sci., 70, 15-25, 2014*
- *Zotter P., Herich H., Gysel M., El-Haddad I., Zhang Y., Mocnik G., Hüglin C., Baltensperger U., Szidat S. and Prévôt A.S.H.: Evaluaton of the absorption Ångström exponents for traffic and wood burning in the Aethalometer-based source apportionment using radiocarbon measurements of ambient aerosol, Atmos. Chem. Phys., 17, 4229-4249, 2017*

---

## Author Response (AR2)

**MS No.: acp-2019-123**

**Title:** *Exploiting multi-wavelength aerosol absorption coefficients in a multi-time resolution source apportionment study to retrieve source-dependent absorption parameters*

Authors: Alice C. Forello et al.

**Response to Reviewers**

In addition to reviewers, the authors acknowledge very much the co-Editor for his suggestions. Comments from Referee #1 were also taken into account in the final version of the paper. The text was carefully checked and revised.

**Co-Editor Decision: Publish subject to minor revisions (review by editor)** (03 Aug 2019)
Comments to the Author:

Dear authors, Thank you very much for your revised manuscript.

The reviewer and I are generally satisfied with your changes. However, there are still many language and grammar issues in the current version. Therefore, the manuscript needs another thorough editorial read (preferable by a native speaker). Although it is not the work of the editor, I marked a few issues below (but not all of them due to limitations in time).

In addition, there are some other minor content issues which need clarification or improvement.

• Line 22: It should be "resolutions".

• Line 28: Please remove the '…' (you already say 'e.g.' at the beginning of the parenthesis).

• Line 32: Add 'the' before 'atmosphere'.

• Line 34 and 36: Alpha and MAC are already defined in Line 19 and 20.

• Line 38: Suggest to move 'approach' behind 'here presented'.

• Line 46: Suggest to replace 'At the state of the art' with 'Currently' or 'As the state of the art'.

• Line 49: 'proved' -> 'proven'. There should probably be a comma after 'In the late 1990s'.

• Line 64: The variable of the absorption coefficient was already defined in line 58. Please add 'to be' before 'proportional'.

• Line 76, line 81 and later: The word 'source apportionment optical models' sounds like a weird construct that is probably not correct. Strictly speaking, it is a model dealing with source apportionment and not with optics. Could you maybe rephrase it as e.g. "Source apportionment models based on optical measurements" or "Source apportionment models using optical measurements as input"?

• Line 82: 'fuels' -> 'fuel'

• Line 84: Add 'the' before 'atmosphere'.

• Line 85: 'need a priori assumption about alpha values' -> 'need a priori assumptions on the alpha values'

• Line 94: This sentence is very long a hard to read. In addition, I would replace 'In the frame' by 'In the framework'.

• Line 95: Also this sentence is difficult to understand. I guess you want to say "Instead of using alpha as an a priori input, this approach even allows to retrieve alpha as a source-dependent value.'?

• Line 102 (and throughout the manuscript): The word 'that' should not be preceded by a comma.

• Line 103: Please remove the '…'.

• Line 104: Alpha has already been introduced many times before.

• Paragraph starting in line 106: This paragraph is a bit confusing and I suggest to rephrase it.

• Line 121: 'in activity' -> 'active'

• Line 178: 'Jülich Forschungszentrum' -> 'Forschungszentrum Jülich'

• The citation of 'Valentini et al.' is not an accepted or peer-reviewed publication and as such should be avoided as reference. I suggest to remove these sentences. It is sufficient that the comparison of the instrument is mentioned in the reply to the reviewers.

• Line 248 and 263: Add 'such' before 'as'.

• Line 265: What do you mean with under-weighed? Less weighed? How?

• Line 297: Suggest to replace 'Opposite' with 'In contrary'.

• Line 304: Add 'the' before 'factor'.

• Line 310: Alpha has been defined already many times before.

• Figure 1: Please add proper y-axis labels (e.g. 'Concentration' or 'Concentration Fe').

• Figure 2: Please add y-axis labels to the unit.

• Line 346: I suggest to add 'the' or 'a' before 'multi-time resolution model'

• Figure 3: It would be helpful to the reader if you describe 'F' and 'EVF' once more in the caption.

• Line 446: The reference to figure S3 is missing in this paragraph.

• Figure 4: As in the right axis, please also add the word 'concentration' or 'conc.' to the y-axis label in the left axis.

• Figure 5: The number on the y-axis for the biomass burning component are somehow wrong. Please check. The font size of the axis numbers is very small (it should be similar to the caption font size).

• Line 511: Alpha was already defined several times before.

• Line 539: Suggest to put a hyphen between alpha and value.

• Line 560: 'At' -> 'For' and add a comma after 'nm'.

• Line 561: Suggest to change it to 'report a MAC value of …'.

• Line 581: Suggest to add a 'the' or 'a' before 'Multiliniear'.

• Line 604: Remove the word 'works'.

• Figure S2: What does "scaled residuals" mean? Normalized by the mean/median? Please clarify in the caption. The y- and x-axis are also missing their proper labels and units.

• Figure S2: Suggest to also mention the year.

Please also consider and reply to the comments by reviewer #1.

*Answer: all comments and changes were implemented in the final version of the paper*

**REFEREE #1**

The authors have responded well to the many points raised in the initial reviews. In particular, the inclusion of far more detail on the methods, and justifications for some of the procedures and assumptions are very welcome. There remain just a few minor points which could be improved.

(a) Lines 512-516 – the wording here is a little confusing. It seems to imply that the Delta-C method is not based on a 2-source hypothesis, whereas clearly it is. It may not have been the intention of the authors to imply this, but to refer to the Delta-C parameter as an input variable in receptor models. Some clarification would be beneficial.

*Answer: the sentence has been changed as follows "Another literature approach used Delta-C as an input variable together with chemical aerosol components in source apportionment models and was very effective in separating traffic (especially diesel) emissions from biomass combustion emissions (Wang et al., 2011, 2012)."*

(b) When discussing the factor associated with sulphate, the authors seek to explain the presence of EC within this factor in relation to the optical absorption, lines 497-501. It is worth bearing in mind that many receptor modelling studies have shown some presence of EC and trace elements in a secondary sulphate factor. This is most probably explicable by the fact that both EC and trace metals are emitted by many sources which also emit sulphur dioxide, for example, fuel oil combustion. Subsequent conversion of sulphur dioxide to sulphate will tend to lead to sulphate condensation on the primary emitted particles, hence leading to a more complex composition for this factor.

*Answer: the sentence has been changed as follows "Opposite to sulphates, EC has a primary origin; however, its presence with a very similar percentage (4-5 %) in a sulphate chemical profile was previously pointed out in Milan, indicating a more complex mixing between primary and secondary sources (Amato et al., 2016) e.g. with sulphate condensation on primary emitted particles. The sulphate factor accounted for 21 % of the PM10 mass."*

(c) It is stated that during the monitoring campaign no significant contribution from mineral dust was observed (lines 549-550). This is expressed in the context of aerosol absorption. However, lines 505 to 508 attribute absorption by resuspended dust to the likely presence of iron minerals. If the term mineral dust is taken to refer to crustal materials and other largely inorganic particles such as those from construction works, then there is ample evidence of the presence of mineral dusts. The statement that there was no significant contribution from mineral dust may have been meant to refer to sources of pure crustal dust such as Saharan dust emissions, but greater clarity is needed here as the text currently appears contradictory.

[revised manuscript text omitted]